# Optimizing spatial organization of FtsZ rings for large-scale constriction in synthetic cells

Anastasija Panevska[1,3,4], Aleksandra Šakanović [1,4], Gianfranco Paccione [2], Germán Rivas[2] & Petra Schwille [1] ✉

Spatially regulated membrane constriction is an important milestone in reconstituting minimal cell division. In giant lipid vesicles, bottom-up approaches have reproduced the assembly, mid-cell positioning, and the initial constriction of an FtsZ-based minimal divisome. However, progressive deformation towards giant vesicle scission by near-equatorial Z rings could so far never be observed. One obvious major limitation has been the scale mismatch, as pure reconstituted FtsZ rings typically exhibit bacterial diameters, too small to constrict typical cell-sized vesicles. Therefore, we explore the potential of other key divisome factors to scale up FtsZ-ring functionality in vitro to match the dimensions required for synthetic cell division. We here focus on cytoFtsN, the cytosolic domain of FtsN, and its effect on FtsZ self-organization. Remarkably, a molar excess of cytoFtsN promotes the formation of large, closed equatorial FtsZ rings on giant vesicle membranes, which are able to constrict to almost full closure. By fluorescence imaging and biochemical analysis, we show that cytoFtsN regulates the spatial organization of the FtsZ network primarily by aligning FtsZ filaments while reducing filament depolymerization. Our findings help to define key requirements in a minimal filament-based system for progressive membrane constriction and thus represent a major step forward towards constructing synthetic cells capable of self-division.

Building a minimal synthetic cell that controls its own division is one of the substantial challenges in bottom-up synthetic biology. Cell division requires the coordinated activity of multiple proteins, especially cytoskeletal components, in concert with their lipid environment, to initiate membrane remodeling and large-scale transformation[1–6]. Despite significant advancements in assembling minimal synthetic cells and reconstituting functional division machineries, achieving a controlled and progressive constriction within simplified synthetic systems remains a significant hurdle[7,8]. Overcoming this challenge would lay a crucial foundation for integrating additional characteristics of life, such as metabolism and gene regulation, thereby significantly advancing the construction of autonomous synthetic cells.

Previous efforts to reconstitute cell division have drawn significant inspiration from the well-characterized and highly advanced model organism, *Escherichia coli*, where cell division is orchestrated by the so-called divisome. This is a highly dynamic multiprotein complex centered around FtsZ, a conserved tubulin homolog and GTPase[9], which polymerizes into filaments that assemble into a ring-like structure, the Z-ring, positioned at the future division site[10–13]. A key feature of FtsZ polymer dynamics is treadmilling, a process in which subunits are continuously added at end of the filament and removed from the

[1]Department of Cellular and Molecular Biophysics, Max Planck Institute for Biochemistry, Martinsried, Munich, Germany. [2]Centro de Investigaciones Biológicas Margarita Salas, Consejo Superior de Investigaciones Científicas (CSIC), Madrid, Spain. [3]Present address: Department of Biology, Biotechnical faculty, University of Ljubljana, Ljubljana, Slovenia. [4]These authors contributed equally: Anastasija Panevska, Aleksandra Šakanović. ✉e-mail: schwille@biochem.mpg.de

other, generating directional movement of FtsZ filaments[14]. FtsZ treadmilling is a GTP hydrolysis-driven process in which filaments continuously polymerize at one end and depolymerize at the other through a cooperative conformational switch, producing directional, circumferential motion around the division site that organizes and guides the movement of peptidoglycan synthases FtsW and FtsI to ensure uniform septal wall synthesis during cytokines[15–17]. It is affected by several factors, including the concentrations of FtsZ and GTP, as well as interactions with other proteins that regulate FtsZ polymerization and stability[18–20]. To ensure proper membrane localization and organization, FtsZ filaments are anchored to the cellular membrane by adapter proteins, such as FtsA, an actin homolog in *E. coli*[21]. Additionally, proteins such as the Min system and SlmA spatially regulate Z-ring dynamics, preventing aberrant division events[22–25]. Furthermore, FtsZ self-organizes into higher-order structures, such as bundles, sheets, and toroids, which are thought to contribute to Z-ring stability and force generation during cell division[26]. The formation of higher-ordered FtsZ-structures is influenced by both the intrinsic properties of FtsZ, and the presence of accessory proteins.

Recent studies have identified a transmembrane divisome protein FtsN as a key activator of bacterial cell division, which coordinates intracellular division events with cell wall remodeling[27]. FtsN functions together with its SPOR-domain partner DamX, which contributes to stabilizing the constriction zone during the late stages of cytokinesis[28]. Through its cytoplasmatic tail (cytoFtsN), FtsN directly interacts with FtsA in the cytoplasm, influencing its oligomeric state[29,30]. In vitro and structural studies have provided detailed insight into how cytoFtsN modulates FtsA and FtsZ dynamics at the onset of division. In reconstituted systems, FtsA couples FtsZ treadmilling to the membrane and recruits cytoFtsN, which is captured by FtsA and co-migrates with FtsZ-FtsA filaments, demonstrating a direct cytoplasmic connection between the early FtsZ-ring scaffold and downstream division factors[30,31]. Structural analyses further revealed that binding of cytoFtsN to FtsA induces a conformational transition from single to antiparallel double filaments, increasing curvature sensitivity and stabilizing the proto-ring on negatively curved membranes[32]. Together, these findings establish cytoFtsN as an allosteric regulator that reorganizes FtsA filaments to strengthen their coupling with FtsZ, thereby priming the divisome for the initiation of constriction.

Most previous attempts to reconstitute the dynamics of the FtsZ, aiming to achieve progressive constriction, have predominantly focused on implementing protein machineries that spatially regulate FtsZ self-assembly and protofilament targeting to the mid-cell membrane[8,33]. Furthermore, the need for macromolecular crowders and crosslinking proteins such as ZapA and ZapD, which stabilize FtsZ filaments and promote bundle formation, has been identified[8,34–36], underscoring the critical role of FtsZ filament organization in efficient cell division. In contrast, an engineered FtsZ chimera containing a membrane-targeting sequence (mts), which enables membrane binding in the absence of native FtsZ anchors ZipA or FtsA, was shown to accumulate at the constricted regions of tubular liposomes (<2.5 μm), thereby laying the foundation for the minimal divisome model[33]. However, FtsZ-mts reconstituted within giant unilamellar vesicles (>5 μm in diameter) forms mini-rings and patches that generate only local, concave distortions without inducing large-scale vesicle constriction[37,38]. On the contrary, co-reconstitution of FtsZ with the gain-of-function FtsA* in giant vesicles produced dynamic Z-rings capable of constriction and occasional septation, whereas substitution of GTP with slowly hydrolyzing GMPCPP abolished ring formation[37]. The in situ expression of FtsA and FtsZ within liposomes resulted in their assembly into curved filaments and closed rings, which constricted membranes into extended necks and budding vesicles[39,40]. Incorporating the Min system further enabled dynamic FtsZ patterning and equatorial ring positioning, though only slight deformations were

achieved[8,41]. Together, these advances charted a progressive path toward a minimal synthetic divisome, yet none captured a continuous, time-resolved transformation driven solely by membrane-tethered FtsZ, without auxiliary anchors, in which a single, stably positioned ring progressively reshapes a giant vesicle into two nearly separated daughter compartments. A key question with regard to building a minimal divisome remains unresolved: how could the diameter of an FtsZ ring be scaled up to effectively constrict giant vesicles while maintaining its function in the division machinery? It has been widely suspected that the dynamic assembly of a more complete divisome would be required to achieve large-scale constriction of synthetic cells[8]. However, while biologically relevant, constructing a synthetic cell with an increasing number of components and experimental factors, along with their inherently complex dynamics, introduces significant challenges[8,22]. Therefore, we decided to radically minimize the number of functional modules simultaneously reconstituted, but aim to identify optimal conditions and features that may enable a reconstituted FtsZ-based divisome to achieve self-positioning and gradual, progressive constriction of a lipid membrane in vitro.

Our study utilizes a purified FtsZ variant with a membrane-targeting sequence (FtsZ-366-mts), the short peptide cytoFtsN, and giant unilamellar vesicles to mimic a minimal synthetic cell. In this highly simplified system, we demonstrate that clearly discernible FtsZ rings are able to progressively constrict across giant vesicles, suggesting a simplified mechanism for large-scale membrane transformation, as required for synthetic cell division. However, the observed membrane constriction depends critically on the prior formation of closed, coherent, large-scale FtsZ-366-mts rings, the assembly of which is regulated by cytoFtsN. Intriguingly, our experiments reveal that cytoFtsN reduces the GTPase activity of FtsZ and lowers the turnover rate of FtsZ monomers, thereby slowing down the treadmilling dynamics. While higher monomer turnover might be expected to increase constriction efficiency, scaling coordinated function to the cellular level appears to require a precise balance between filament turnover and spatial organization. In summary, our results define the minimal requirements for significant in vitro membrane constriction through FtsZ-based rings, and provide an experimental demonstration even without spatial regulators such as the Min proteins. This represents a significant milestone for the engineering of self-dividing synthetic cells from minimal functional modules.

## Results

### CytoFtsN stabilizes FtsZ bundles on the surface of the vesicles or reconstituted within the vesicles

Given the proposed role of FtsN as a key activator of bacterial cytokinesis[27], we considered it a promising candidate for reconstituting a minimal synthetic cell division machinery. Rather than using full-length transmembrane FtsN, we focused on its cytoplasmic tail, cytoFtsN, which is responsible for the interaction with intracellular cytoskeletal elements[29,30]. We initially analyzed FtsZ assemblies in the presence of cytoFtsN on lipid vesicles using two different assays (Fig.1). First, we examined wild-type FtsZ assembly on the inner and outer surfaces of GUVs, in the presence of FtsA, the native FtsZ anchor (Fig. 1a). Second, to bypass the need for FtsA, we employed an FtsZ variant (FtsZ-366-mts) fused to an amphiphilic helix, referred to as membrane targeting sequence (mts), that allows direct FtsZ attachment to the membrane (Fig. 1b) [18]. The construct lacks the C-terminal region responsible for the interactions of FtsZ with other proteins[38], enabling us to assess the minimal requirements for FtsZ-ring constriction independently of additional cellular factors. For visualization, wild-type FtsZ was labeled with Alexa488, while the FtsZ-366-mts chimera was genetically fused with Venus, a variant of yellow fluorescent protein, as previously done[8,38].

To investigate the interplay between cytoFtsN and FtsZ-366-mts, we reconstituted the cytoskeletal elements within lipid vesicles

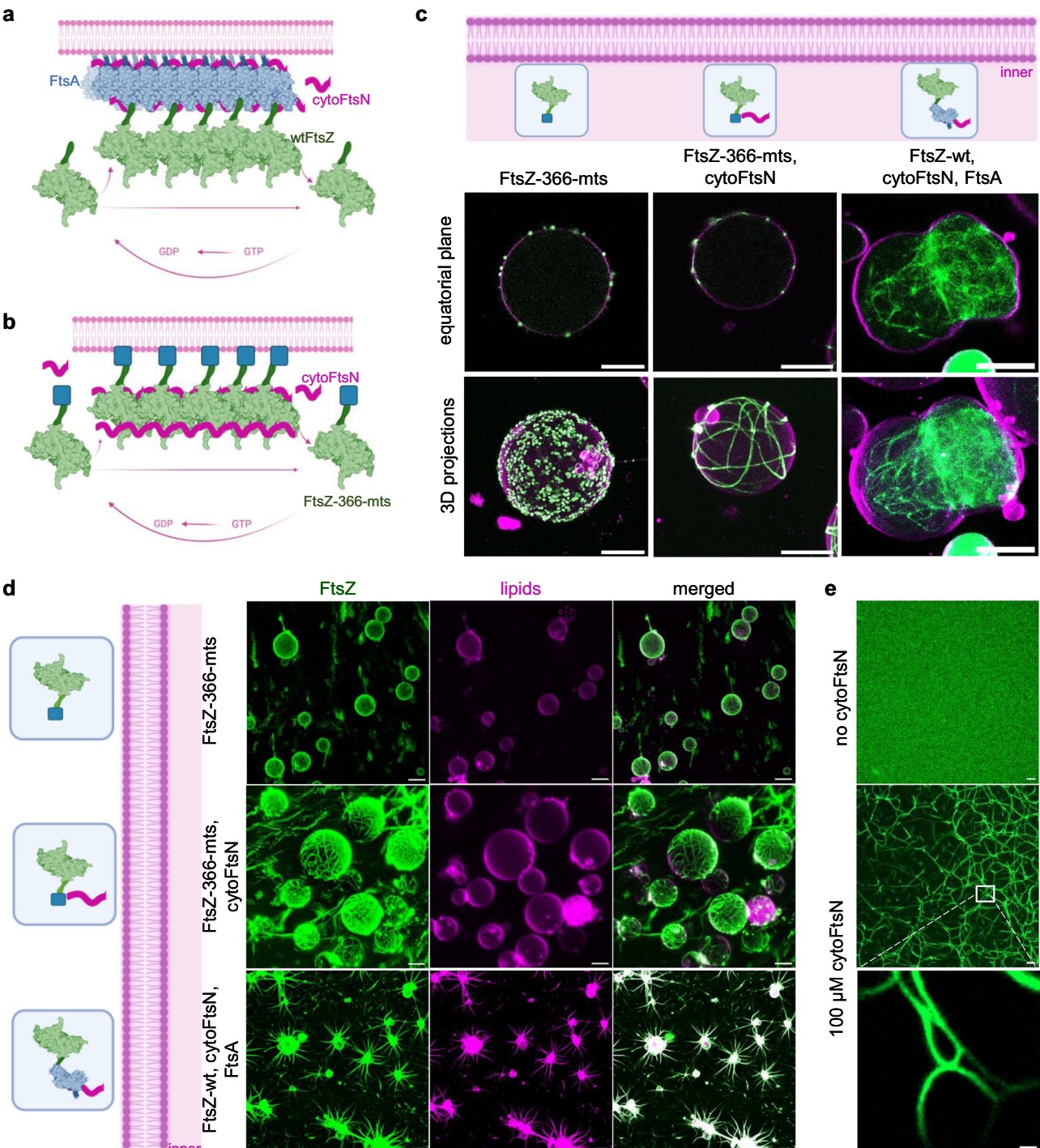

**Fig. 1 | CytoFtsN enhances FtsZ bundle formation inside and on the surface of lipid vesicles. a** Schematic representing the native proteins required for early-stage division ring assembly. **b** Identified building blocks for constructing a minimal division system. **c** Representative confocal images of FtsZ-366-mts reconstituted in lipid vesicles ( ± cytoFtsN), and wild-type (wt) FtsZ membrane-tethered via FtsA with cytoFtsN. Top: 2D equatorial images showing FtsZ-366-mts (green) binding to the lipid membrane (magenta), with 100 μM cytoFtsN, and wt FtsZ (green) with FtsA and 100 μM cytoFtsN. Bottom: 3D projections showing FtsZ-366-mts mini-rings in the absence of cytoFtsN, and pronounced bundles in the presence of cytoFtsN, for both FtsZ-366-mts and wt FtsZ. Experimental conditions: 5 μM FtsZ-366-mts, 0.5 mM GTP, 1 mM GMPCPP (left panel), with 100 μM cytoFtsN (middle panel), and 10 μM wt FtsZ, 2.5 μM FtsA, 100 μM cytoFtsN, 0.5 mM GTP, 1 mM GMPCPP, 1 mM ATP (right panel). Scale bar: 10 μm. **d** FtsZ organization on the surface of lipid vesicles as wt FtsZ anchored via FtsA and as FtsZ-366-mts variant, in the presence or absence of cytoFtsN. 3D projection of FtsZ (green, FtsZ-366-mts/Alexa488 or wt

FtsZ, left panel), membrane (magenta in the middle panel) and merged images (right panel) are presented. FtsZ-366-mts filament bundles induced by GTP (top) show increased density upon cytoFtsN addition (middle). When cytoFtsN is co-added with FtsA and wt FtsZ, FtsZ organization is substantially altered, leading to pronounced membrane deformation and formation of lipid tubular structures. Experimental conditions: 5 μM FtsZ-366-mts, 1 mM GTP (top panel); same as top panel with 100 μM cytoFtsN (middle panel); 5 μM wt FtsZ, 1.25 μM FtsA, 100 μM cytoFtsN, 1 mM GTP, 1 mM ATP (bottom panel). Scale bar: 10 μm. **e** Assemblies of the FtsZ-366-mts in solution, without cytoFtsN (no cytoFtsN), where FtsZ remains diffusely distributed, and upon addition of cytoFtsN (100 μM cytoFtsN). Experimental conditions: 5 μM FtsZ-366-mts, 1 mM GTP (top panel), same as top panel with 100 μM cytoFtsN (middle and bottom panel). Scale bars: 10 μm (overview), 1 μm (higher magnification). Schematic illustrations were created in BioRender. Panevska, A. (2026) https://BioRender.com/4ersjae.

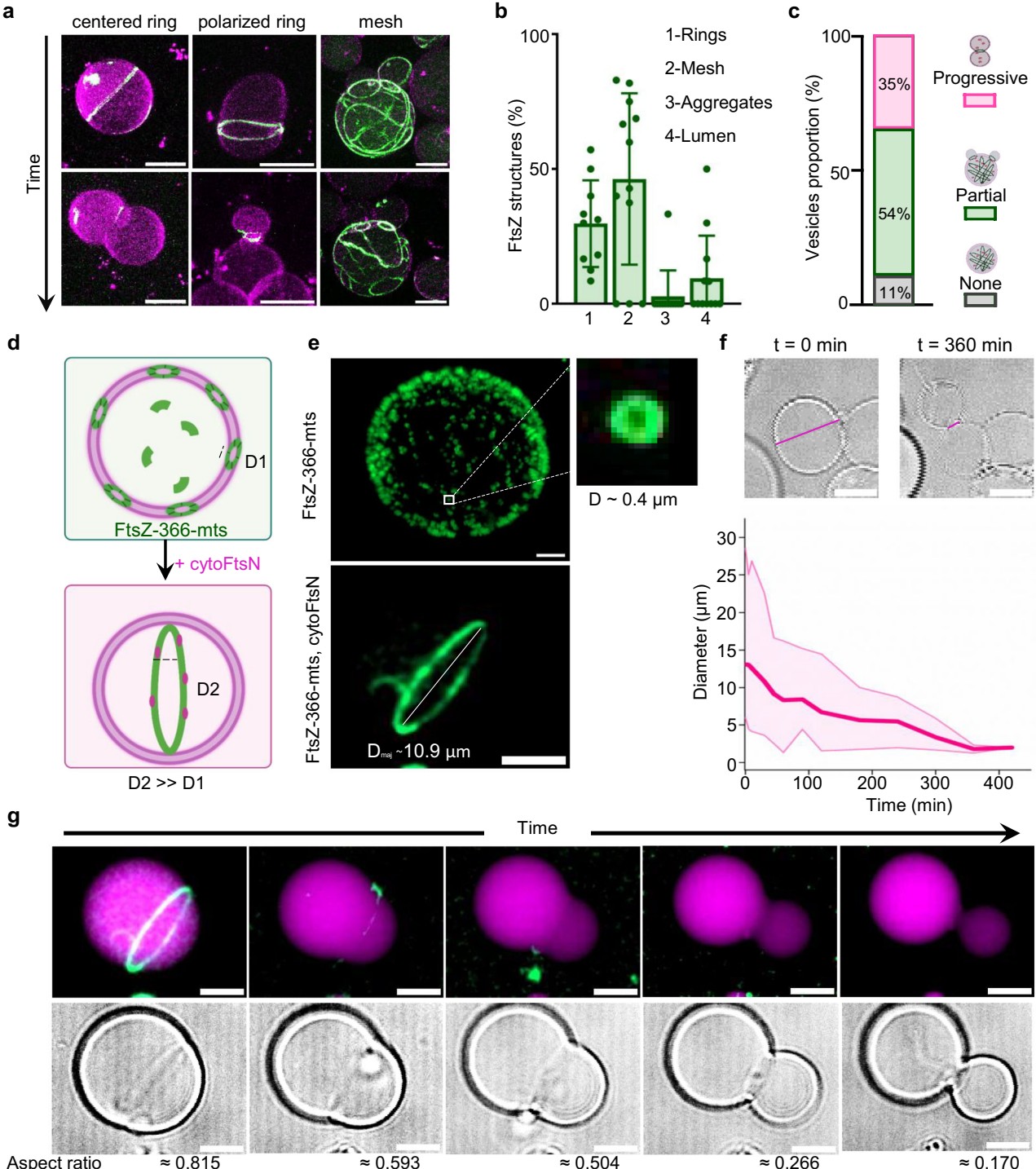

(Fig. 1c). In the absence of cytoFtsN, FtsZ-366-mts localized to the inner membrane leaflet as discrete, mini-ring structures (Fig. 1c, left panel), consistent with previous observations of membrane-anchored FtsZ[38]. However, the addition of a 20-fold molar excess of cytoFtsN, determined empirically, dramatically altered FtsZ-366-mts organization, transforming mini-rings into an extended network of membrane-bound bundles (Fig. 1c, middle panel). Similarly, when wild-type FtsZ was combined with cytoFtsN, bundles also formed. However, these structures were found within the vesicle lumen, as wild-type FtsZ alone is unable to bind the membrane (Supplementary Fig. 1a, right panel). This observation underscores a previously unrecognized role of cytoFtsN in modulating FtsZ assembly and filament organization. Introducing FtsA as an anchor allowed FtsZ to localize to the

membrane, where the bundles formed a mesh-like structure (Fig. 1c, right panel), with cytoFtsN colocalizing with Alexa488-labeled wild-type FtsZ (Supplementary Fig. 1b). Although membrane deformations were observed as a consequence of FtsZ/FtsA/cytoFtsN bundle formation, they did not yield aspect ratios of deformed vesicles below 0.7 (Fig. 1c, right panel).

To further evaluate the effect of cytoFtsN on FtsZ network organization, we added it either externally to GUVs or tested it in solution, independently of lipid membranes. Expectedly, the cytoFtsN significantly affected the organization of the FtsZ filament network (Fig. 1d, e, Supplementary Fig. 1c). In contrast to the membrane-bound bundles formed by FtsZ-366-mts alone (Fig. 1d, top panel), the addition of cytoFtsN resulted in visibly straighter and more uniformly aligned

**Fig. 2 | FtsZ/cytoFtsN ring drives progressive vesicle deformation.**
**a** Representative 3D projections of vesicles containing FtsZ-366-mts structures formed in the presence of 100 μM cytoFtsN (*n* > 90 vesicles). Upper panel: Vesicles contained full, closed rings, localized either on mid-cell (centered ring) or at the poles (polarized ring), or a mesh of FtsZ-366-mts/cytoFtsN bundles (mesh). Bottom panel: vesicle shown on the upper panel after 2–6 h. Bottom panel: vesicle shown on the upper panel after 2–6 h. **b** Distribution of FtsZ structures observed within the lipid vesicles 30 min after initiation of the experiment. Representative 3D projections for these categories are presented in (**a**). **c** Percentage distribution of vesicles displaying protein-driven membrane constriction, either partial or progressive. **d** Schematic illustration: cytoFtsN-promoted assembly of large-scale FtsZ-ring. Created in BioRender. Panevska, A. (2026) https://BioRender.com/zvhqt5r. **e** 3D projections of 5 μM FtsZ-366-mts (green) encapsulated in lipid vesicles. In the absence of cytoFtsN, FtsZ forms mini-rings with an apparent diameter of approximately 0.4 μm (upper panel). In the presence of cytoFtsN, a representative

large FtsZ-ring is visible at the equatorial plane (D = 10.9 μm). The apparent elliptical shape results from the ring being tilted relative to the imaging plane. Scale bar: 10 μm. **f** Brightfield images of the equatorial plane showing the diameter reduction in a lipid vesicle over 6 h (upper panel). Scale bar: 10 μm. Diameter of GUVs (*n* = 14) measured in the equatorial plane from brightfield images over time (bottom panel). The magenta line represents the mean diameter, with light pink shading indicating the minimum–maximum range. **g** Time-lapse images of the FtsZ-366-mts/cytoFtsN ring and protein-driven vesicle constriction. Upper panel: 3D projections of the merged image of FtsZ-366-mts (green) and ATTO655-cytoFtsN (magenta) showing a contractile ring that progressively deforms vesicles into an asymmetric dumbbell shape. After 2 h FtsZ-366-mts signal disappeared due to photobleaching. Bottom panel: Brightfield images of the equatorial plane of the vesicles. Experimental conditions on all presented panels were kept constant: 5 μM FtsZ-366-mts and 100 μM cytoFtsN with 0.5 mM GTP and 1 mM GMPCPP. Scale bar: 5 μm. Source data for this figure is available in the Source Data file.

filaments (Fig. 1d, middle panel). On the other hand, when wild-type FtsZ was anchored with FtsA, and cytoFtsN was added to GUVs, we observed significant membrane remodeling and extensive liposome tubulation (Fig. 1d, bottom panel, Supplementary Movie 1). The observed large-scale membrane deformations are consistent with the known membrane-remodeling activity of the FtsA[40]. However, the presence of cytoFtsN notably amplified membrane deformation, likely by directly interacting with FtsA as shown previously and promoting enhanced FtsZ localization on the membrane surface[30]. These observations suggest that cytoFtsN influences FtsZ behavior, both indirectly through FtsA and via direct interaction with FtsZ, resulting in distinct effects on membrane remodeling. Importantly, by comparing FtsZ structures formed in solution with and without cytoFtsN, we confirmed that the stabilizing effect of 100 μM cytoFtsN on FtsZ bundles occurs independently of the lipid membrane (Fig. 1e). Additionally, we observed colocalization of cytoFtsN with FtsZ bundles in solution, indicating direct interaction between these two elements (Supplementary Fig. 1c). Taken together, by identifying the cytoFtsN effect in promoting the formation of FtsZ bundles, the FtsZ-366-mts/cytoFtsN system emerged as a promising candidate for a minimal synthetic cell division machinery.

### FtsZ-366-mts/cytoFtsN act as a minimal protein-based machinery for progressive FtsZ-ring constriction and membrane deformation

To characterize the FtsZ/cytoFtsN system within vesicles, we fine-tuned the experimental conditions to manipulate FtsZ filament morphology and ideally obtain well-defined rings (Fig. 2). Specifically, to promote FtsZ polymerization and to stabilize the resulting structures, we used both GTP and the slowly hydrolysable GTP analog GMPCPP[7]. When GTP alone was used to induce FtsZ polymerization in the presence of cytoFtsN, the resulting FtsZ bundles led to protein aggregation within an hour, due to rapid polymerization and instability of the Z-ring. In contrast, the combination of GMPCPP and GTP together with cytoFtsN resulted in enhanced polymerization and FtsZ-ring stability, producing longer-lived, aligned and less diffusive FtsZ polymers. Previous reconstitution studies combining FtsZ, FtsA and Min proteins often relied on molecular crowding agents to enhance FtsZ bundling and facilitate the formation and equatorial positioning of large rings in giant vesicles[8], though such agents were also reported to hinder efficient constriction[33,42]. Notably, crowders are not universally required, as confinement in narrow geometries can also support functional small-scale ring assembly[33,42]. Here, we sought to stabilize FtsZ bundles and division rings using only cytoFtsN, without the need for crowding agents. In the presence of Min protein pole-to-pole oscillations, we achieved transient FtsZ/cytoFtsN-ring positioning to the vesicle center. However, since FtsZ/cytoFtsN-rings were mostly self-positioning close to the equatorial region of the vesicle even without Min proteins, by

critically increasing the bending modulus of the bundles, we proceeded using only FtsZ and cytoFtsN and therefore avoided adding complexity to the system.

Lipid vesicles encapsulating FtsZ-366-mts and cytoFtsN, with GTP and GMPCPP, were visualized approximately 15 min after preparation. We observed vesicles containing different FtsZ structures, either well-defined, closed FtsZ rings or a mesh of FtsZ bundles (Fig. 2a). Quantitative analysis revealed that while the majority of vesicles contained the mesh of FtsZ bundles, a significant fraction of vesicles carried fully-formed FtsZ rings (Fig. 2b). Notably, the FtsZ structures remained dynamic, undergoing significant changes over time (Fig. 2a). The mesh of bundles remained stable for around 2–3 h, after which protein aggregates began to accumulate in the vesicle lumen. In contrast, vesicles containing closed FtsZ rings exhibited pronounced morphological transitions, characterized by extensive lipid membrane remodeling and the formation of furrow-like membrane invaginations in concert with ring constriction (Fig. 2a, f, g). To quantitatively assess membrane deformation, we measured the aspect ratio (length/width) of constricting vesicles over time. Based on aspect ratio changes, we categorized membrane constrictions into two groups, namely partial, if the aspect ratio remained above 0.6, and progressive, if the aspect ratio went below 0.25 (Fig. 2c). A key requirement for progressive constriction was the formation of a single fully closed FtsZ ring, distinct from the previously described reconstituted diffuse Z rings that lack structural coherence[8]. Importantly, encapsulation of both FtsZ-366-mts and cytoFtsN led to an increase in ring diameter (D = 10.9 μm), which coincided with large-scale membrane deformation in the micrometer range (Fig. 2d–g). In contrast, encapsulation of FtsZ-366-mts alone resulted in the formation of significantly smaller rings with an estimated apparent diameter of ~0.4 μm, which is near the diffraction limit of our imaging system, and these structures failed to induce membrane deformations beyond the initial ring dimensions (Fig. 2e). This suggests that both, organization and sizes of the FtsZ rings are critical parameters determining the scale and progression of membrane constriction.

The progressive constriction of the single, fully closed FtsZ/cytoFtsN ring was visually captured by time-lapse confocal imaging of the gradual evolution of vesicle shape from sphere to dumbbell over six hours (Fig. 2f, g, Supplementary Movies 2, 3). This represents a clear demonstration that a single, continuous, membrane-tethered FtsZ ring can accomplish large-scale transformation of a spherical giant vesicle to almost full fission, where the constriction process could be followed in real time. Importantly, progressive equatorial constriction could be observed in every vesicle where a single fully closed FtsZ/cytoFtsN ring had been formed, regardless of vesicle size (Fig. 2f, Supplementary Fig. 2). This observation reinforces the importance of closed-ring formation for large-scale membrane transformation, consistent with previous theoretical models[43].

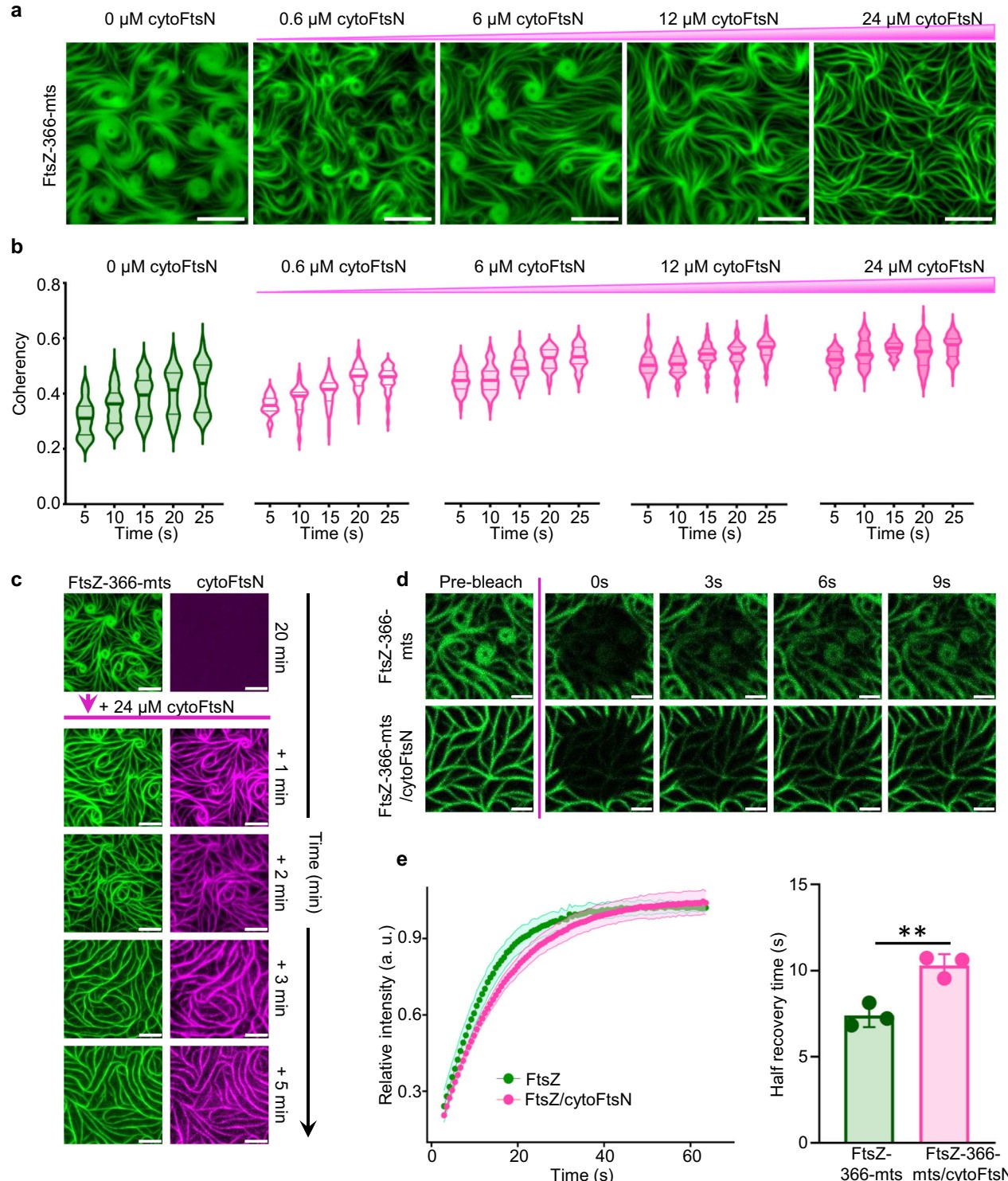

## CytoFtsN enhances the architectural coherence of the FtsZ-366-mts network

Functionally comprehend and quantitatively analyze the effect of cytoFtsN on the FtsZ structural architecture, we turned to supported lipid bilayer (SLB) assays and high-resolution TIRF microscopy. Using SLBs as a biomimetic platform, membrane-attached FtsZ filaments can be studied with higher spatiotemporal precision[30,31]. Consistent with previous studies, we observed that FtsZ filaments, when tethered to SLBs, self-organize into networks characterized by highly curved architectures with the presence of swirling vortices[18,20,44] (Fig. 3a, left). We next examined the FtsZ architecture in the presence of increasing, defined concentrations of cytoFtsN. The addition of cytoFtsN to FtsZ-mts in equimolar concentration (both 0.6 μM) did not significantly change the FtsZ filaments architecture. On the contrary, at higher concentrations, cytoFtsN induced notable reorganization of the FtsZ network and altered its architecture, leading to the disappearance of curved, swirling vortices typical for FtsZ (Fig. 3a). Specifically, FtsZ filaments transitioned into more aligned and straightened bundle configurations at a 20-fold molar excess of cytoFtsN (12 μM). The architectural transformation exhibited a maximal effect at a 40-fold molar excess of cytoFtsN, where swirling vortices were completely absent (Fig. 3a). To quantitatively assess the impact of cytoFtsN on the

**Fig. 3 | CytoFtsN reshapes the FtsZ filament network on supported lipid bilayers. a** Representative micrograph ($n > 10$) showing architecture of 0.6 μM FtsZ-Venus-mts filament patterns (green) formed in the absence and presence of varying concentrations of the cytoFtsN, 10 min after the addition of 0.8 mM GTP. The concentrations of cytoFtsN were 0 μM, 0.6 μM, 6 μM, 12 μM, and 24 μM, respectively. **b** Distribution of mean coherency values of the FtsZ filaments in the absence of cytoFtsN (0 cytoFtsN) and in the presence of increasing concentration of cytoFtsN (0.6 μM, 6 μM, 12 μM, and 24 μM) at five selected time points taken from a time-lapse experiment. Data were obtained from three independently repeated experiments per condition. For each time point, three images, each from a separate independent replicate, were analyzed. Each image was subdivided into 25 regions of interest (ROIs) for quantification. **c** CytoFtsN promptly reorganizes steady-state FtsZ-mts filament patterns. Sequential snapshots capturing FtsZ filament morphology in steady state (20 min after GTP-induced polymerization) and at subsequent time points (1 min, 2 min, 3 min, and 5 min) following addition of 24 μM

cytoFtsN. **d** Representative snapshots of the fluorescence recovery after photo-bleaching (FRAP) of FtsZ in the absence (top) and presence of 24 μM cytoFtsN (bottom). **e** Half-time recovery was affected by cytoFtsN addition. Data (left) shows an average normalized fluorescence intensity with shaded bars (SD) from three independently repeated experiments, where in each experiment between 8 and 12 regions of interest were bleached, while using between 3 and 5 unbleached regions for normalization. Data represent mean ± SD from $n = 3$ independent experiments. Half recovery time (right), calculated for each repetition ($n = 3$) and presented as single data bar plots (mean ± SD), demonstrating the cytoFtsN effect on filaments turnover rate. *$p < 0.05$. Unpaired two-tailed t-test results revealed a significant difference ($p = 0.0058$). Concentrations of FtsZ-Venus-mts and GTP in all presented experiments were kept constant (0.6 μM and 0.8 mM GTP). Scale bar: 5 μm at panels a and c, 2 μm at panel d. Source data for this figure is available in the Source Data file.

---

FtsZ architecture reconstituted on SLBs, we analyzed the dominant local orientation of the FtsZ structures using a structure tensor model[45]. We performed orientation analysis within a defined, Gaussian-shaped local window of approximately 0.2 μm. Within the local window, the image gradient was determined as a 2D vector, describing how the intensity changes and in what direction. We visualized local orientation and local coherency analysis by generating color maps and coherency maps, both of which show striking differences in the organization of the FtsZ filament network depending on the presence and concentration of cytoFtsN (Supplementary Fig. 3a, b). Next, we quantified the architectural organization of FtsZ network using the coherency maps generated from selected time points in the time-lapse experiments, where pixel values range from 0 (for isotropic regions) to 1 (indication perfect local alignment) (Fig. 3b). In the absence of cytoFtsN (0 uM cytoFtsN), FtsZ filament networks displayed bimodal distribution of coherency values and progressive increase in overall order over time, with the median coherency rising from 0.31 at 5 min to 0.44 at 25 min. The addition of cytoFtsN at equimolar concentrations (0.6 μM) did not significantly alter the median coherency values compared to FtsZ alone. In contrast, higher concentrations of cytoFtsN (≥6 μM) resulted in an increase in network alignment, reflected by significantly elevated coherency values. Interestingly, higher isotropy was already apparent at the earliest time points, with the medial coherency values of 0.52 at 5 min to 0.58 at 25 min, suggesting that cytoFtsN induces a rapid reorganization effect on the FtsZ filament network.

To determine whether cytoFtsN colocalizes with membrane-anchored FtsZ filaments, we performed dual-color TIRF microscopy using ATTO 655-labeled cytoFtsN[30,44] (Fig. 3c, Supplementary Fig. 3c, d). Fluorescence intensity profiles revealed substantial colocalization between labeled cytoFtsN and FtsZ filaments (Supplementary Fig. 3c). The spatial correlation of the fluorescence signals persisted throughout the observation period. Furthermore, time-resolved imaging of cytoFtsN introduced to pre-formed, steady-state FtsZ networks confirmed rapid bundle stabilization, occurring within the first minute of cytoFtsN addition (Fig. 3c). The immediate structural response suggests a robust regulatory capacity of cytoFtsN in modulating FtsZ bundle organization.

To investigate the impact of cytoFtsN on FtsZ filament dynamics, we performed fluorescence recovery after bleaching (FRAP) experiments in the presence and absence of cytoFtsN (Fig. 3d, e). The addition of cytoFtsN induced mode prolongation in FRAP recovery half-time, indicating that while subunit exchange remains active, it occurs at a slower rate. The slower turnover of FtsZ subunits induced by cytoFtsN likely reflects the increased size and stability of the filaments. Altogether, our findings demonstrate that cytoFtsN organizes filaments into bundles and modulates FtsZ subunit dynamics, resulting in a stabilized filament architecture with reduced subunit exchange kinetics.

## CytoFtsN affects FtsZ network organization in ionic strength-dependent way, while it decreases GTPase activity

To gain a better understanding of how cytoFtsN may affect FtsZ filament architecture, we examined the electrostatic compatibility between cytoFtsN and wild-type FtsZ. FtsZ is predominantly negatively charged, while cytoFtsN carries a positive charge (Fig. 4a), suggesting an electrostatically driven interaction. To quantify the binding affinity of cytoFtsN and FtsZ in solution, we conducted fluorescence anisotropy measurements of labeled cytoFtsN at ionic strengths lower than physiological levels (100 mM KCl) and at higher ionic strength (500 mM KCl). We determined a low-affinity interaction (Kd ≈ 10.5 μM) for low salt levels, which was weakened nearly tenfold at higher ionic strength, indicating an electrostatically driven association (Fig. 4b).

Next, we investigated the effect of cytoFtsN on FtsZ polymers using fluorescence anisotropy (Fig. 4c). All experiments were conducted in the presence of GTP to ensure the polymerization of FtsZ filaments. Increasing concentrations of cytoFtsN led to a rise in initial anisotropy values, indicating a reduction in the rotational mobility of labeled FtsZ, which suggests the formation of larger, more stable FtsZ assemblies. The effect of increased initial anisotropy was reversed under higher ionic strength conditions, further supporting the notion of an electrostatic interaction. Dynamic light scattering (DLS) measurements revealed a similar trend, showing a substantial increase in light scattering at the higher cytoFtsN concentration (Fig. 4f). While DLS cannot precisely determine the size of the FtsZ bundles due to their polydisperse nature, the increase in scattered light indicates a cytoFtsN-induced increase in the average polymer size in a concentration-dependent manner. Notably, both fluorescence anisotropy and DLS demonstrated that the bundling effect of cytoFtsN is decreased at higher ionic strength.

TIRF measurements provided additional insights into the ionic-strength-dependent effects of cytoFtsN on FtsZ filament dynamics. Near the physiological range (150 mM KCl), FtsZ filaments without cytoFtsN formed dynamic swirling vortices (Figs. 2a, 4d). At higher ionic strength conditions (300 mM KCl), the diameter of vortices was smaller, and the bundles appeared less interconnected, confirming that ionic strength modulates filament dynamics[46]. Higher-salt (300 mM KCl) significantly reduces the above-described effect of cytoFtsN on filament organization, leading to filament arrangement similar to that observed in the absence of cytoFtsN (Fig. 4d).

To further evaluate the ionic nature of the interaction between cytoFtsN and FtsZ, we examined previously reported cytoFtsN variants containing substitutions within the conserved, positively charged RRKK motif (residues 16–19)[29,30]. Specifically, the RRKK motif was replaced with either negatively charged (DDEE) or charge-reduced/small hydrophobic (RAAK) residues. Using TIRF microscopy and the same experimental conditions as for wild-type cytoFtsN, including a 40-fold molar excess of the peptides, we observed that neither peptide variant affected the FtsZ patterns (Fig. 4e). Specifically, the FtsZ-mts

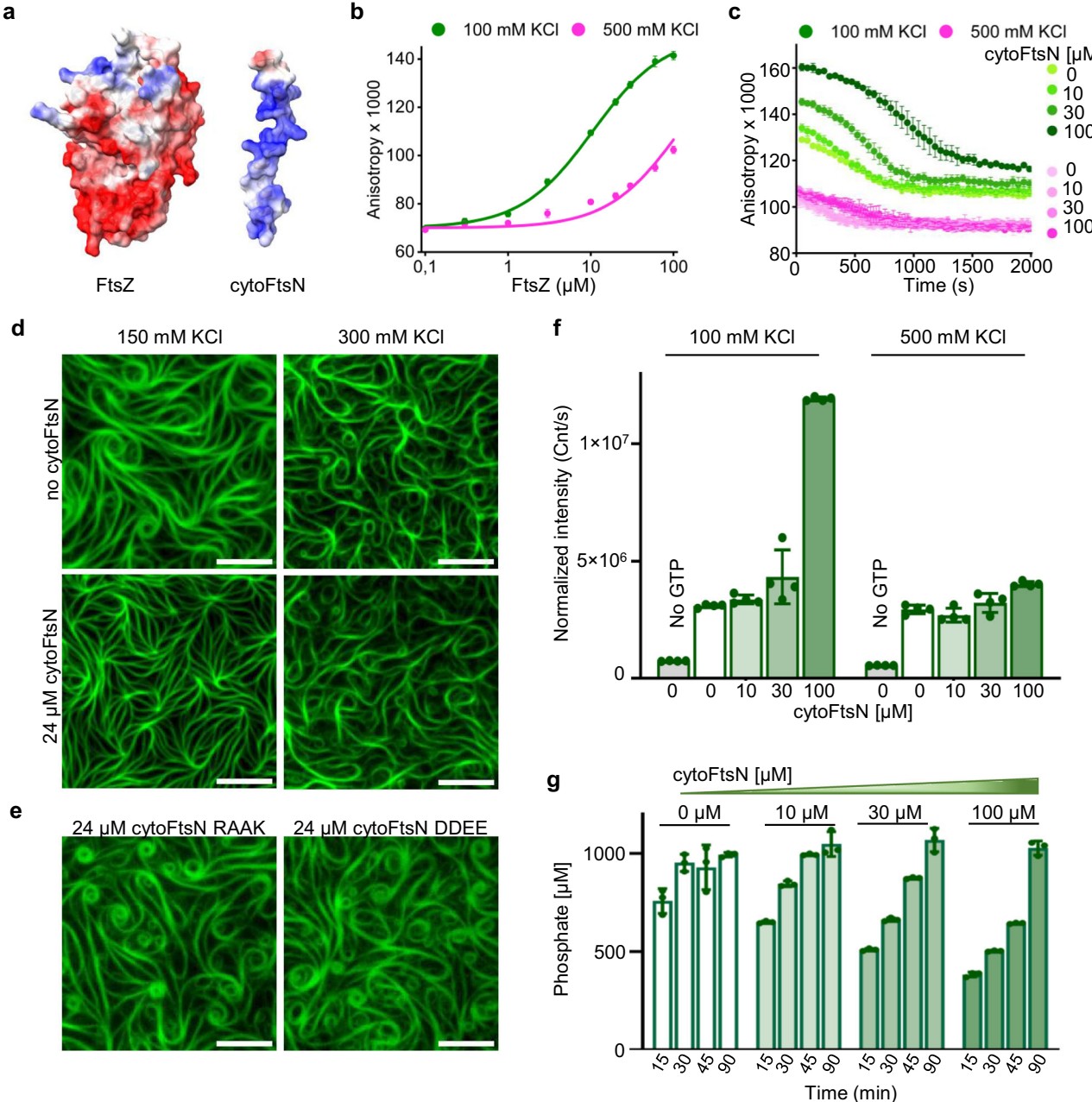

**Fig. 4 | CytoFtsN induced FtsZ bundling in solution and on membranes at varying ionic strengths. a** Surface electrostatic potential of wtFtsZ (PDB ID: 6UMK EcFtsZ (L178E)-GTP[37]) and of the predicted cytoFtsN structure. **b** Fluorescent anisotropy of 0.1 μM FtsN-655 binding to wt FtsZ (0.1–100 μM) in buffer containing either 100 mM KCl (green) or 500 mM KCl (magenta). Data represent mean values ± SD from *n* = 3 independently repeated experiments. **c** Depolymerization kinetics of 10 μM FtsZ labeled with Alexa-488 in the presence of 1 mM GTP under varying cytoFtsN and KCl concentrations. In 100 mM KCl (green circles), FtsZ filaments undergo continuous depolymerization over time due to GTP hydrolysis, with different shades representing the indicated cytoFtsN concentrations. In 500 mM KCl (magenta circles), the kinetics are altered, reflecting the impact of higher ionic strength on filament stability by cytoFtsN. Data represent mean values ± SD from *n* = 3 independently repeated experiments. **d** Representative TIRF microscopy (from *n* = 3 independent experiments) showing the spatial organization of FtsZ filaments (0.6 μM) on SLBs under varying ionic conditions. Upper: Filament architecture at 150 mM KCl (left) and 300 mM KCl (right) without cytoFtsN (no

cytoFtsN). Lower: Filament organization upon addition of 24 μM cytoFtsN. Scale bars: 5 μm. **e** Representative TIRF images (from *n* = 3 independent experiments) showing organization of the FtsZ-mts filaments in the presence of 24 μM cytoFtsN variants, in which the conserved RRKK motif (residues 16–19) was substituted with either charge-reduced/hydrophobic (RAAK) (left) or negatively charged residues (DDEE) (right). Scale bar: 5 μm. **f** Dynamic light scattering (DLS) measurements of 10 μM FtsZ at 100 mM or 500 mM KCl and varying cytoFtsN concentrations. Normalized DLS signals indicate cytoFtsN-induced filament bundling, which is diminished at higher ionic strength. Data represent mean values ± SD from *n* = 4 independently repeated experiments. **g** GTPase assay of 10 μM FtsZ with 1 mM GTP in 100 mM KCl buffer at varying cytoFtsN concentrations. The data show a cytoFtsN concentration-dependent decrease in phosphate released over time, indicating reduced GTP hydrolysis upon cytoFtsN binding. Data represent mean values ± SD from *n* = 3 independently repeated experiments. Source data for this figure is available in the Source Data file.

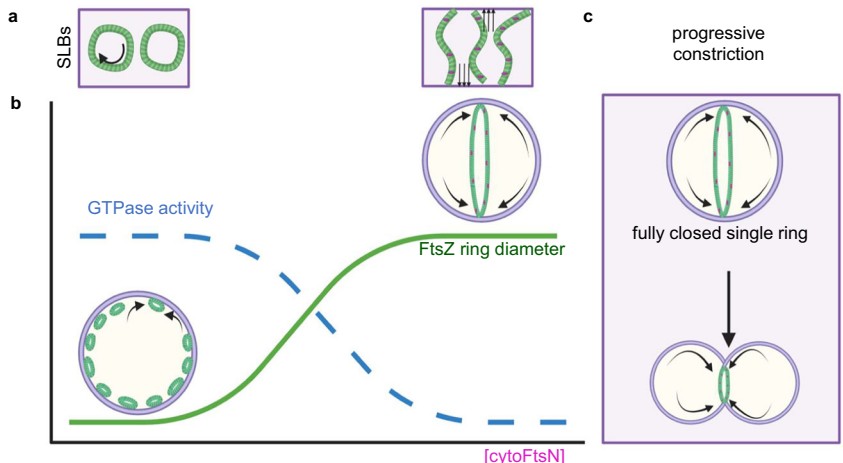

**Fig. 5 | Model for a single continuous FtsZ ring, stabilized by cytoFtsN, as the minimal architecture for large-scale vesicle constriction. a** On supported lipid bilayers, FtsZ filaments self-organize into dynamic, curved vortices. The addition of cytoFtsN in molar excess triggers a structural transition from circular patterns to more aligned and straightened bundles. **b** When encapsulated within giant vesicles in the absence of cytoFtsN, FtsZ organizes into mini-rings, too small to encircle the vesicle. The presence of excess cytoFtsN facilitates the assembly of a single, fully closed ring matching the GUV diameter. This rearrangement is driven primarily by non-specific electrostatic interactions between cytoFtsN and FtsZ, which also lead to a reduction in the GTPase activity of FtsZ. This reduced GTP turnover stabilizes filament contacts, favoring parallel alignment over curved vortex-like patterns. The formation of scaled-up rings appears to be a direct consequence of bundle straightening filaments that cannot bend strongly, but instead adopt the vesicle's curvatures and align along their circumferences. **c** The architecture of the FtsZ network is critical for large-scale constriction. Rather than a mesh or multiple overlapping rings, a single continuous ring, preferably positioned close to the equator, but also functional when non-centrally positioned, acts as the minimal structural unit capable of driving deformation. Once established, this ring supports progressive constriction of the giant lipid vesicle, leading to marked deformation, however, without full scission. The reduced filament dynamics in the presence of cytoFtsN likely prolong the lifetime of the ring, allowing sufficient time for vesicle shape change. Created in BioRender. Panevska, A. (2026) https://BioRender.com/r7mp29j.

structural reorganization and bundling effect was not detected in the presence of either the charge-inverted (DDEE) or the charge-reduced (RAAK) variant. These findings, together with the sensitivity of the interaction between cytoFtsN and FtsZ to ionic strength (Fig. 4b–f), suggest that surface charge compatibility (Fig. 4a), rather than specific sequence recognition, is likely the key driver of the interaction. Furthermore, structural prediction using AlphaFold3 gives a low confidence in modeling a stable complex between cytoFtsN and FtsZ (Supplementary Fig. 4), consistent with transient or predominantly nonspecific electrostatic interactions.

Finally, we quantified the effect of cytoFtsN on FtsZ GTPase activity by measuring the inorganic phosphate released over time (Fig. 4g). CytoFtsN decreases the GTPase activity of FtsZ in a concentration-dependent manner, with the highest concentration of cytoFtsN tested reducing the GTPase activity by at least 50%. Complete GTP hydrolysis by FtsZ required 90 min in the presence of cytoFtsN, compared to 30 min in the absence of cytoFtsN. The reduced GTPase activity correlates with reduced FtsZ depolymerization dynamics observed by anisotropy measurements (Fig. 4c).

Together, our results demonstrate that excess cytoFtsN decreases FtsZ GTPase activity and promotes the reorganization of FtsZ networks from curved vortices on SLBs or multiple mini-rings in GUVs into stabilized, large-scale rings that span vesicle dimensions (Fig. 5). This FtsZ architecture is sufficient to support progressive membrane constriction. Thus, a cytoFtsN-stabilized, vesicle-spanning FtsZ ring defines the minimal structural unit required for large-scale protein-driven deformation of giant lipid vesicles.

## Discussion

Division of a synthetic cell requires large-scale constriction and scission of its lipid membrane, a process ideally regulated and guided by a membrane-remodeling cytoskeletal ring. In this study, we report the successful in vitro reconstitution of progressive membrane constriction by a minimal division ring composed of the bacterial cytoskeletal protein FtsZ and a cationic peptide fragment, cytoFtsN. We show that, through a careful orchestration of structural proteins and environmental components, such as ionic strength and cross-linking agents, complex functionalities, including contractile rings, can be reconstituted with a minimal set of modules. Such simplified systems promise to be a versatile platform for integrating additional biomolecular machinery to replicate various aspects of cellular life.

FtsZ plays a central role in driving membrane invagination, yet its mechanical sufficiency in isolation has been debated. While pioneering studies established that FtsZ-Venus-mts alone can assemble rings to constrict narrow tubular liposome ( < 2.5 μm), its inability to constrict larger GUVs, forming only local distortions, suggested a requirement for auxiliary anchors like FtsA or ZipA[33]. Indeed, co-reconstituting FtsZ with the gain-of-function FtsA*, or expressing wild-type FtsA and FtsZ in cell-free systems, yielded dynamic rings and budding necks[37,38,40]. These FtsA-FtsZ assemblies frequently formed clusters that constricted the membrane into budding necks, yet under the studied conditions, they were insufficient to complete abscission. In contrast, by utilizing FtsZ-Venus-mts and cytoFtsN, we demonstrate here that FtsZ-mts alone is sufficient to drive global constriction in giant vesicles. This overcomes limitations observed in previous multicomponent systems, such as the cell-free expression of FtsA, FtsZ, and MinCDE, which resulted in equatorially centered but diffuse rings capable of only shallow membrane indentations[8]. Additionally, the interplay between FtsZ, FtsA, and membrane-associated proteins was reported to induce membrane deformation in the form of lipid tubulation[31,32,47]. In our system, the simultaneous presence of FtsZ, FtsA, and cytoFtsN on lipid vesicles resulted in extensive lipid tubulation and vesicle collapse (Fig. 1d), likely due to enhanced FtsZ recruitment and its subsequent effects on the lipid membrane. Specifically, cytoFtsN appeared to potentiate the formation of FtsA double filaments, which in turn likely increased the membrane-associated local density of FtsZ, leading to a surface crowding effect that drove large-scale membrane deformations. In the absence of FtsA, however, cytoFtsN promoted the

formation of robust FtsZ-mts and FtsZ-366-mts filament bundles, revealing that its stabilizing effect on FtsZ is independent of FtsA (Fig. 1, Supplementary Fig. 1). Although the cytoplasmic tail of FtsN is known to interact with FtsA, direct interactions between cytoFtsN and FtsZ had not been previously recognized.

Previous studies have established that MinCDE pole-to-pole oscillations are essential for guiding FtsZ-ring formation and positioning by driving the condensation of a diffuse, mesh-like network of FtsZ filaments into discrete ring-like structures at the equatorial region of vesicles[8,41]. However, our findings here reveal that equatorial ring formation of FtsZ-366-mts can occur even in the absence of MinCDE-mediated spatial organization, e.g., if cytoFtsN induces enhanced bending rigidity of large-scale FtsZ bundles in the form of closed rings, which are then self-targeted to regions of maximum circumference (Fig.2, Supplementary Fig. 2). Intriguingly, cytoFtsN-mediated stabilization produced a significant population of vesicles with persistent, equatorially localized and closed FtsZ rings, thereby being independent of MinCDE-based positioning. Although further refinement may be needed to improve the precision of ring positioning, our system represents a significant advancement towards the reconstitution of stable simplified FtsZ-ring formation in synthetic cells.

A key outcome of our study is the discovery that FtsZ-366-mts rings, being stabilized with cytoFtsN, induce pronounced and progressive giant vesicle deformation, captured here for the first time (Fig. 2). In contrast to previous studies[8,37,39–41], our system represents a conceptual and methodological advance in reconstituting FtsZ-ring driven constriction at the scale of giant vesicles. Rather than relying on multiple recombinant proteins such as FtsA or its hyperactive variant FtsA*, or on cell-free expression systems requiring continuous protein synthesis, we used a minimal two-component setup consisting of FtsZ-366-mts and a short cytoplasmic peptide of FtsN. Our findings demonstrate that large-scale constriction of giant vesicles can be achieved by a single FtsZ ring structure alone, when combined with a regulatory peptide that enhances its dynamic self-organization. This minimal configuration eliminates the need for accessory membrane anchors while preserving functional coupling between FtsZ dynamics and membrane remodeling. Remarkably, under these conditions, we observed large-scale constriction of giant vesicles ($>2.5\,\mu m$ in diameter) initiated from a single, centrally positioned FtsZ ring that progressively tightened over time until near-complete vesicle division, capturing the full transformation process rather than only its intermediates or budded final stages. The use of cytoFtsN as a small, diffusible modulator provides a strategy for tuning filament alignment, stability, and constriction dynamics without requiring stoichiometric assembly of larger accessory proteins, thereby significantly simplifying and stabilizing the minimal divisome reconstitution. Our findings highlight the intrinsic membrane-transformative capacity of FtsZ. Obviously, further components may be necessary to execute the final step of membrane scission on the molecular level, although this has previously been accomplished by peripheral membrane-curvature-inducing agents in general. By establishing a simplified yet functional synthetic division system, we provide insights into the intrinsic capabilities of FtsZ to induce substantial membrane remodeling when appropriately stabilized.

We hypothesize that the main role of cytoFtsN, rather than promoting or actively generating constrictive force, is to reinforce filament architecture, enhancing network cohesion and persistence (Fig.3a, b). Our biochemical characterization (Fig. 4) further suggests that the interaction between FtsZ and cytoFtsN is mainly electrostatic. The cationic cytoFtsN likely neutralizes the negative charges on FtsZ monomers, thereby reducing electrostatic inter-filament repulsion and promoting tighter filament packing. This structural stabilization correlates with a decrease in FtsZ's GTPase activity (Fig. 4), consistent with increased filament longevity and resulting in larger closed rings. Additionally, testing cytoFtsN variants in which conserved positively charged RRKK was replaced by negatively charged (DDEE) or charge-reduced (RAAK) residues showed that neither variant induced structural reorganization of FtsZ filaments (Fig. 4e), highlighting the role of peptide charge. FtsZ is not unique in its ability to form bundles through electrostatic interactions or the action of crosslinking/bundling proteins. Indeed, cation interactions, encompassing both non-specific electrostatic effects and specific ion binding, have been demonstrated to promote the bundling of actin filaments[48]. In the case of actin, bundling is driven by a reduction in electrostatic repulsion between filaments once a threshold cation concentration, necessary for actin polymerization, is exceeded[48]. For instance, high concentrations of divalent cations have been shown to condense actin filaments into bundles and induce their over-twisting, resulting in an increase in bundle bending persistence length[48]. Analogously, we propose that cytoFtsN, acting as a cationic agent, interacts non-specifically with negatively charged FtsZ filaments, promoting bundle formation and altering their structural properties.

Together, our findings establish a robust minimal biochemical framework for membrane constriction by a minimal cytoskeletal protein ring in a synthetic system (Fig. 5). First, the use of a membrane-anchored FtsZ variant (FtsZ-366-mts) ensures stable association with the vesicle surface, providing a scaffold for ring formation. Second, the addition of GMPCPP in the correct ratio with GTP decreases FtsZ dynamics, while the addition of cytoFtsN enhances FtsZ filament packing and stabilization, highlighting its central role in promoting filament cohesion. Third, the architecture of the FtsZ ring is critical: effective constriction requires a well-aligned, continuous, and uniformly organized ring with a width of up to -0.5 μm, rather than an irregular or diffuse ring-like structure. Once such a ring is established, the stabilization of its constituent filaments is essential for the system to reorganize and progressively constrict over a many-micrometer scale. In our system, this stability is achieved through a two-step process: first, by globally reducing filament turnover with the addition of GTP analog GMPCPP together with GTP, and second, by the architectural action of cytoFtsN, which constrains the filaments into a robust, higher-order structure. This behavior contrasts with the small-scale divisome observed in bacterial cells, emphasizing the importance of kinetic and architectural regulation for engineering division rings in enlarged, synthetic compartments. We hypothesize that the stabilized FtsZ-cytoFtsN ring then constricts through a finely-tuned combination of both filament shortening and turnover, triggered by the gradual depletion of GTP within the vesicle. Our results significantly advance the broader objective of developing synthetic cells capable of autonomous division while deepening our understanding of bacterial cytokinesis at a fundamental level.

## Methods

### Protein purification

Protein Purification/ppFtsZ-366-mts was purified by ammonium precipitation as previously described[18]. Briefly, the protein was expressed in *E. coli* BL21(DE3)pLysS from a pET-11b vector overnight at 20 °C. Following cell lysis by sonication and clarification via centrifugation, the supernatant was treated with 30% ammonium sulfate for 20 min on ice. The resulting precipitate was collected by centrifugation, resuspended, and purified via anion exchange chromatography using a 5 × 5 ml Hi-Trap Q-Sepharose column (GE Healthcare). Wild-type FtsZ was isolated through two cycles of calcium-induced precipitation and purified by anion exchange chromatography using a 5 ml Hi-Trap Q-Sepharose column[49]. The column was equilibrated in Tris-glycerol buffer (containing 5 mM MgCl₂), and the protein was eluted with a 0–1 M KCl gradient, with peak fractions typically occurring at 500–600 mM KCl. Purified proteins were dialyzed into Tris-250 KCl buffer (containing 10% glycerol). Wild-type FtsZ was subsequently labeled at the N-terminus with Alexa Fluor 488 carboxylic acid succinimidyl ester (Thermo Fisher Scientific)[50]. Protein concentrations

were determined using the Bradford Assay (Bio-Rad, Hercules, CA, USA). Protein aliquots were flash-frozen in liquid nitrogen and stored at −80 °C.

## Peptide synthesis

The wild-type cytoplasmic peptide of FtsN (cytoFtsN), with the sequence H-MAQRDYVRRSQPAPSRRKKSTSRKKQRNLPAV-OH, was synthesized in two versions: untagged and labeled with ATTO655 fluorescent dye at the N-terminal. CytoFtsN variants were conserved RRKK motif (residues 16–19) was substituted with either negatively charged (DDEE, sequence MAQRD YVRRS QPAPS DDEES TSRKK QRNLP AV-OH) or charge-reduced/hydrophobic (RAAK, sequence MAQRD YVRRS QPAPS RAAKS TSRKK QRNLP AV-OH) residues, were synthetized as untagged and unlabeled peptides. All variants were synthesized using solid-phase peptide synthesis following standard Fmoc chemistry protocols. The crude peptides were purified to >95% homogeneity using reverse-phase HPLC, and their identity was confirmed by mass spectrometry. For mass spectrometry analysis, fractions obtained after purification were analyzed to confirm peptide synthesis. Specifically, five fractions per injection were examined, with 60 mg of synthesized peptide loaded per injection. Following analysis, the relevant fractions were pooled and freeze-dried to obtain the final peptide sample. Mass spectrometry analysis was performed on four independent peptide synthesis replicates for cytoFtsN ($n = 4$) and on a single replicate for the variants DDEE and RAAK peptides. No additional biological controls were included, as the analysis was performed for analytical confirmation of peptide identity. Mass spectrometry analysis was performed using an Agilent 1100 coupled to a microTOF mass spectrometer (Bruker Daltonik). Peptide samples (3 μL) were injected at 10 °C and eluted under isocratic conditions at 70 % buffer B for 3 min at a flow rate of 250 μL/min. The eluents consisted of Buffer A (0.05 % TFA in $H_2O$, pH 2.0) and Buffer B (0.05 % trifluoroacetic acid (TFA) in acetonitrile (ACN), pH 2.0). The microTOF mass spectrometer (Bruker Daltonik) was operated in positive ion mode over an m/z range of 400–2000, and MS1 scans were recorded. Data processing was conducted using Compass™ 1.7 DataAnalysis 4.2 software (Bruker Daltonik). Spectra were deconvoluted using the MaximumEntropy algorithm with an instrument resolving power of 10,000. Lyophilized peptides were dissolved at a concentration of 10 mg/mL in Binding Buffer (50 mM HEPES-KOH, 100 mM potassium acetate, 5 mM magnesium acetate, pH 7.7) and stored in small aliquots at 4 °C.

## Lipid vesicle preparation

Depending on the experimental conditions, the giant unilamellar vesicles were prepared either by the double emulsion transfer method[51] or electroformation[52]. Briefly, to encapsulate proteins, the vesicles were prepared by double emulsion transfer using a lipid mixture consisting of 1-palmitoyl-2-oleoyl-glycero-3-phosphocholine (POPC) and 1-palmitoyl-2-oleoyl-sn-glycero-3-phospho-(1′-rac-glycerol) (POPG) (Avanti Polar Lipids, Alabaster, AL, USA) at an 8:2 molar ratio. For membrane visualization, 0.1 mol % of ATTO655-labeled 1,2-dioleoyl-sn-glycero-3-phosphoethanolamine (DOPE) (ATTO-Tech GmbH, Siegen, Germany) was added to the lipid mixture. A 50 μL lipid mixture (25 mg/ml) was dried under a nitrogen stream, and the dried lipids were dissolved in 10 μL of decane (TCI Deutschland GmbH, Eschborn, Germany), followed by the addition of 500 μL of mineral oil (Carl Roth GmbH, Karlsruhe, Germany). The inner aqueous solution, containing 5 μM FtsZ-366-mts, 100 μM cytoFtsN or cytoFtsN-ATTO655, 10 mg/mL BSA, 0.5 mM GTP, and 1 mM GMPCPP in reaction buffer (50 mM HEPES−KOH, 100 mM ($CH_3COO$)K and 5 mM ($CH_3COO$)$_2$Mg, pH 7.7) was prepared. The emulsion was created by combining 5 μL of the inner solution and 250 μL of the lipid-oil mixture. Separately, 500 μL of the outer solution was added to a fresh tube, followed by a 200 μL layer of the lipid-oil mixture to form a lipid monolayer. The emulsion was then layered onto a multi-layered solution. Centrifugation at

6000 x $g$ for 15 min at 22 °C resulted in the formation of GUVs. The oil phase and supernatant were removed, and the remaining -100 μL containing the GUVs was resuspended. A 50 μL aliquot was transferred to a fresh tube for imaging. To examine the effects of FtsZ and cytoFtsN on the surface of membrane vesicles, GUVs were prepared by electroformation in PTFE chambers with platinum electrodes. Briefly, 6 μL of the lipid mixture (2 mg/mL in chloroform) composed of POPC and POPG in a molar ratio 8:2 with 0.1 mol% Atto655-DOPE was spread onto the platinum wires and dried in a desiccator for 30 min. The chamber was then filled with 350 μL of an iso-osmolar sucrose solution, and an AC electric field (2 V (RMS), 10 Hz for 1 h, followed by 2 Hz for 0.5 h) was applied at 20 °C. After electroformation, the chamber was cooled to room temperature (22–24 °C) before experiments.

## 3D confocal analysis of FtsZ structures and vesicle shape transformation

Confocal microscopy was performed using a Zeiss LSM900 laser scanning microscope equipped with a C-Apochromat 40x/1.20 water-immersion objective (Carl Zeiss AG). Excitation wavelengths of 488 nm (for Alexa488 and Venus) and 633 nm (for ATTO655) were used. Images (512 × 512 pixels) of GUVs with reconstituted proteins were acquired with Z-stacks (40–120 slices, 0.21–0.48 μm intervals) at 2–5 min intervals for durations ranging from 30 min to 10 h. Image processing, visualization, and analysis were performed using Fiji ImageJ 1.54p[53]. Z-stacks were visualized as 3D maximum intensity projections, and FtsZ structures were manually identified from these projections. Representative wide-field images and multiple micrographs demonstrating consistent results were included to ensure data reproducibility. All micrographs presented in the main manuscript and supplementary information represent results from at least three independent replicates.

## Quantitative analysis of vesicle deformation and FtsZ assemblies

To determine the aspect ratio (short axis length / long axis length) of the vesicles containing FtsZ-rings, the diameter of individual GUVs ($n = 14$) was measured over time from brightfield equatorial-plane images acquired at indicated time points. For each vesicle, the diameter was determined using line profiles across the vesicle and converting pixel distance to micrometers based on the microscope calibration. Measurements were repeated for each time point until the vesicle was no longer visible or its size remained unchanged. The status of FtsZ structures and their impact on vesicle morphology were classified into three categories: bundles forming a mesh, bundles forming a full FtsZ ring localized near a pole, inducing membrane deformation, and a mid-cell localized FtsZ full ring, leading to membrane deformation. The frequency of vesicles exhibiting constriction-driven membrane deformation (partial or progressive, without complete scission) upon the formation of a full FtsZ ring at either a polar or mid-cell position was quantified. More than 30 lipid vesicles were analyzed per sample allowed for robust characterization of the observed phenomena. FtsZ-ring diameter was measured by fitting an ellipse to the segmented ring structure in Fiji, with major and minor axis lengths extracted and converted to micrometers based on pixel resolution.

## Preparation of supported lipid bilayers (SLBs)

SLBs were prepared by vesicle fusion from small unilamellar vesicles (SUVs) as previously described[18]. Briefly, a lipid mixture of 1-palmitotl-2-oleoylphosphatidylcholine (POPC) and 1-palmitotl-2-oleoylphosphatidylglycerol (POPG) dissolved in chloroform was combined in a 7:3 molar ratio. For bilayer visualization experiments, DOPE-ATTO 655 (0.01 mol%) was incorporated into the mixture. For attaching the His-tagged cytoFtsN on SLBs, the SLBs were prepared with POPC, POPG and either 1 mol% Ni-NTA or 2.5 mol% Ni-NTA dioleoyl-glycero-succinyl (Ni-NTA DOGS). The solution was dried under nitrogen gas, followed by vacuum desiccation for >1 h to remove

residual solvent. Multilamellar vesicles (MLVs) were generated by hydrating the dried lipid film with reaction buffer (50 mM Tris-HCl, 150 mM KCl, 5 mM MgCl$_2$, pH 7.4) to a final lipid concentration of 4 mg/mL, followed by vigorous agitation for 30 min. Next, SUVs were prepared by bath sonication of MLVs for 15 min. To prepare SLBs, glass coverslips were first cleaned using oxygen plasma treatment (30 s) before attaching silicon chambers (IBID). SUVs solution (150 μL, 0.5 mg/mL total lipids) was added to each chamber, and bilayer formation was initiated by adding CaCl$_2$ (1 mM). After incubation at 37 °C for 10 min, unfused vesicles were removed through 6–10 washing steps using 400 μL of preheated (37 °C) SLB buffer (50 mM Tris-HCl, 150 mM KCl, pH 7.4). The SLB buffer was then gradually exchanged with reaction buffer (50 mM Tris-HCl, 150 mM KCl, 5 mM MgCl$_2$, pH 7.4).

## TIRF microscopy

The FtsZ-filaments patterns were established by preparing 50 μL of a 5 × concentrated solution containing 3 μM FtsZ in reaction buffer (50 mM Tris-HCl, 150 mM KCl, 5 mM MgCl$_2$, pH 7.4), followed by the addition of GTP (4 mM). The solution was then added to SLBs covered with 200 μL reaction buffer (final volume 200 μL). To test the effect of cytoFtsN on FtsZ assembly, the peptide at varying concentrations (3, 30, 60, or 120 μM) was mixed into the 50 μL solution containing FtsZ monomers prior to initiating polymerization with 4 mM GTP. For colocalization experiments, ATTO 655-labeled cytoFtsN was added to pre-formed FtsZ patterns. The cytoFtsN-ATTO655 was introduced at either 12 μM or 24 μM final concentration. The FtsZ patterns were monitored immediately after FtsZ and GTP, with or without cytoFtsN, were added to SLBs. All experiments were conducted using an inverted total internal reflection fluorescence (TIRF) microscope (Elyra 7, Zeiss). Fluorophores were excited using 488 nm and 642 nm lasers. The emitted fluorescence was filtered through a filter set (525/50 nm and 705/75 nm emission filters). For dual-color imaging, we employed a two-track acquisition mode with frame-wise alternation between channels. Fluorescence signals were detected using two EMCCD cameras (Andor iXon 897). Dynamic protein patterns were monitored for either 30 or 60 min with images acquired at 1 s intervals using 15 ms exposure time. All imaging was performed using a 63x oil-immersion TIRF objective (NA = 1.46).

## FtsZ network orientation analysis

The local orientation of FtsZ filament networks was quantified from original, scaled TIRF microscopy images using the OrientationJ plugin, version 2.0.75[55], within Fiji ImageJ 1.54p[54]. For time-series experiments, image frames at defined time points (5 min, 10 min, 15 min, 20 min, and 25 min) were selected. The orientation fields were computed using a structure tensor approach. The image gradient required for the structure tensor was determined using cubic spline interpolation and a Gaussian local window with a sigma (σ) of 2.0 pixels (corresponding to approximately 0.194 μm). The analysis generated Color-coded orientation maps and coherence maps. Coherency maps, where pixel intensity (from 0 to 1) reflects local orientation anisotropy, were used to quantify network order. Mean coherence for each defined region of interest was calculated directly. A total of 25 regions of interest, systematically placed on a grid covering the central area, were analyzed per approximately 50 μm$^2$ frames. Each condition was evaluated in three independently repeated experiments.

## Fluorescence recovery after photobleaching (FRAP)

FRAP experiments were conducted using a Zeiss LSM 980 confocal microscope. For sample preparation, 50 μL of a solution containing 3 μM FtsZ-Venus-mst, with or without 120 μM cytoFtsN, was added to SLBs covered with 200 μL of reaction buffer (50 mM Tris-HCl, 150 mM KCl, 5 mM MgCl$_2$, pH 7.4). SLBs with bound FtsZ-Venus-mts, with or without cytoFtsN, were imaged. Following the addition of GTP

(0.8 mM in the final 250 μL), systems were allowed to equilibrate for 15 min before photobleaching experiments began. Within a 54 μm field of view, 8–12 circular regions of interest (ROIs, 8 μm in diameter) were selected for photobleaching using a high-intensity laser. Post-bleaching fluorescence recovery was monitored by acquiring time-series images at 100 ms intervals over 70 s. To quantify fluorescence recovery, the intensity of the bleached areas was normalized against a non-bleached reference region. The normalized fluorescence intensity was plotted against time, and the half-time recovery (t½) was determined by fitting the data to an exponential recovery function in OriginPro 2023 using the following equation:

$$I(t) = I_{(bl)} + A(1 - e^{-t-TD/\tau}) \tag{1}$$

where I(t) is the fluorescence intensity at time t, $I_{(bl)}$ is the intensity immediately after bleaching, representing the base line, A is the amplitude of recovery, TD represents the time delay before recovery starts and ⊤ is the characteristic recovery time constant. Half-time recovery was calculated as t1/2 = τ ln(2). Statistical analysis was performed using GraphPad Prism 10. All experiments were independently repeated at least three times. To assess the statistically significant difference in the half-time recovery of FtsZ fluorescence in the presence and absence of cytoFtsN, the Shapiro-Wilk test was first performed to evaluate normality of the residuals. Since the Shapiro-Wilk test indicated that the residuals were normally distributed ($p = 0.238$), an unpaired $t$-test was subsequently conducted to determine if there was a significant difference between the two conditions.

## Anisotropy experiments of FtsZ-cytoFtsN

Fluorescent anisotropy measurements were performed in a Spark® Multimode microplate reader (TECAN) at 25 °C with 485 nm excitation and 535 nm emission filters for FtsZ-Alexa 488 depolymerization assays, and 645 nm excitation and 705 nm emission filters for cytoFtsN-ATTO 655 binding experiments. In both cases, a 384 black polystyrene (non-binding), flat-bottom microplate (Corning) was used, measuring the anisotropy values in the final 20 μL reaction volumes. FtsZ depolymerization curves were monitored over time with a constant concentration of FtsZ (10 μM) and a tracer amount of labeled protein, FtsZ-Alexa 488 (50 nM), in the presence of 1 mM GTP and varying concentrations of cytoFtsN (0, 10 μM, 30 μM and 100 μM). Two buffer compositions were tested, a low and a high ionic strength buffer: 50 mM Tris, 100 or 500 mM KCl, 5 mM MgCl$_2$, pH 7.5. For the binding titrations, labeled cytoFtsN-ATTO-655 (50 nM) was incubated with increasing concentrations of FtsZ and the same low and high ionic strength buffers were employed. For data fitting and approximation of the dissociation constant ($K_d$), we used the following equation:

$$r = r_F + \frac{(r_B - r_F) \cdot x}{K_d + x} \tag{2}$$

where $r$ represents the observable anisotropy, $r_F$ and $r_B$ are the anisotropy values for the free and bound cytoFtsN peptide, $K_d$ dissociatyion constant, and $x$ is the total FtsZ concentration. The simplified fitting of the anisotropy binding data is possible since the binding affinity is weak and the $K_d$ is at least 100 times higher than the concentration of ligand used (50 nM cytoFtsN-ATTO655), ensuring that the unbound FtsZ concentration is basically the same as the total FtsZ concentration employed.

## GTPase assay of FtsZ

FtsZ GTPase activity was quantified by measuring the release of inorganic phosphate (Pi) over time using the BIOMOL® GREEN assay (Enzo Life Sciences). Reactions were performed with 10 μM FtsZ, 1 mM GTP, and increasing concentrations of cytoFtsN (0, 10 μM, 30 μM and

100 μM) in reaction buffer (50 mM Tris, 100 mM KCl, 5 mM MgCl$_2$, pH 7.5). Pi was quantified for all samples after 15, 30, 45, 60, and min. Samples were then diluted to a final volume of 50 μL in the reaction buffer prior to reacting with the BIOMOL GREEN to ensure measurements fell within the standard curve range. Finally, the absorbance at 620 nm was measured in a transparent 96-well plate (Greiner) using Spark® Multimode microplate reader (TECAN), following a 20 min incubation with the 100 μL BIOMOL GREEN.

## Dynamic light scattering
Dynamic light scattering (DLS) was measured using the DynaPro Nanostar (Wyatt Technology) to investigate the effect of cytoFtsN on FtsZ filament or bundle formation. Normalized scattering at 658 nm was obtained for 10 μM of FtsZ before and after adding 2 mM GTP, as well as for FtsZ filaments assembled in the presence of increasing cytoFtsN concentrations (0, 10 μM, 30 μM and 100 μM). Each sample was loaded into a disposable Nanostar micro cuvette and measured 15 times, with 4 replicates per condition, ensuring a stable scattering signal for at least 5 min at 20 °C. DLS measurements were conducted using buffers with lower ionic strength (50 mM Tris, 100 mM KCl, 5 mM MgCl$_2$, pH 7.5) and higher ionic strength (50 mM Tris, 500 mM KCl, 5 mM MgCl$_2$, pH 7.5).

## Structural model prediction by AlphaFold3
Structural models of the cytoFtsN–FtsZ complex were generated using AlphaFold3 via the AlphaFold Protein Structure Database. A 1:1 complex between cytoFtsN and FtsZ was modeled, and the five predicted structures, along with their associated confidence metrics (Predicted Aligned Error (PAE) matrices and per-residue predicted Local Distance Difference Test (pLDDT) scores), were retrieved from the online interface. All models were analyzed using the AlphaFold database tools and visualized in ChimeraX v1.9 for figure preparation. When models are colored by AlphaFold3 confidence scores, the color scheme follows the default AlphaFold3 convention: blue (pLDDT > 90, very high confidence), cyan (70–90, high confidence), yellow (50–70, low confidence), and orange (pLDDT < 50, very low confidence).

## Statistics and reproducibility
Sample sizes were chosen based on standards commonly used in the field and on practical considerations related to vesicle yield and imaging throughput. For vesicle-based experiments, analyses were performed on vesicles obtained from at least three independent biological replicates per condition. The total number of vesicles analyzed per experiment ranged from 10 to >30. Microscopy experiments were repeated independently at least three times for each experimental condition, yielding qualitatively and quantitatively consistent results. Representative micrographs shown in the main figures and Supplementary Information were selected from these independent replicates. The experiments were not randomized. The investigators were not blinded to allocation during experiments and outcome assessment.

## Reporting summary
Further information on research design is available in the Nature Portfolio Reporting Summary linked to this article.

## Data availability
The source data generated in this study are provided in this paper. The original microscopy image data are available under restricted access due to their large file size ( >10 GB). These data are freely available by contacting the corresponding author, who will aim to process all requests within one week. Structural data used in this study are available at the PDB accession 6UMK. The protein sequence for the truncated FtsZ-366-mts variant has been deposited in the NCBI GenBank database under accession PX929019. Source data are provided with this paper.

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

## Acknowledgments

We would like to thank the MPIB Bioorganic Chemistry and biophysics core facility for peptide synthesis, the MPIB Protein production core facility for protein purification (RRID: SCR_025739). We acknowledge members of Petra Schwille's lab for helpful discussions, in particular Dr. Adrian Merino and Dr. Shunshi Kohyama, Dr. Jan-Hagen Krohn for the help with TIRF microscopy and Sandra Ortmeier for support in preparation of SLBs. Funding was provided by the European Research Council (ERC Synergy Grant MetaDivide, no. 101167181) to P.S. and A.S., by the Alexander von Humboldt Foundation to A.P., the Spanish Government through Grant PID2019-104544GB-I00 funded by MICIU/AEI/10.13039/501100011033 to G.R. and Agencia Estatal de Investigación of the Spanish Government through Grant PRE2020-092044 to G.P.

## Author contributions

P.S., A.P., and A.Š. conceived the study. A.P. designed, performed, and analyzed all experiments involving lipid vesicles and microscopy assays in solution. A.Š. designed, performed, and analyzed all experiments with supported lipid bilayers (SLBs). G.P. designed, performed, and analyzed the biochemical characterization and solution-based assays. G.R. provided advice on crowding conditions and helpful discussions. A.P. and A.Š. wrote the original draft. All authors contributed to writing and reviewing the final manuscript.

## Funding

## Competing interests

The authors declare no competing interests.
