## [Peer Review File · Nature Communications]

Optimizing spatial organization of FtsZ rings for large-scale constriction in synthetic cells

Corresponding Author: Professor Petra Schwille

Version 0:

Reviewer comments:

Reviewer #1

(Remarks to the Author)

This paper addresses the question of how to reconstitute FtsZ ring assembly and constriction in large spherical liposomes. They use primarily the construct FtsZ-mts, and show for the first time that adding an excess of cytoFtsN, a 32 aa cationic peptide, enhances the assembly of intact Z rings, which can progressively constrict the liposomes. The GUV constriction seems essentially the same as that reported by Godino and Danelon 2022, using a somewhat different combination, FtsZ (native) plus FtsA*. Although the constriction is very similar, the present result is interesting in showing that there is nothing essential about FtsA*. The generic FtsZ-mts can do the job if provided with a cationic peptide, which probably acts by bundling FtsZ protofilaments. That puts a renewed interest in FtsZ itself as a source of constriction force.

However, the presentation has a major flaw. This is epitomized in the statement (Line 228) "To the best of our knowledge, this is the first large-scale protein-based vesicle transformation process observed in real time." Do the authors really not know the history of related studies? Are they really trying to claim that their FtsZ-FtsN system is the first to achieve "large-scale" constriction?

The authors heavily reference their own 2022 Kohyama and Schwille study for prior studies of reconstitution of FtsZ in liposomes, which used crowding and the Min system to improve the assembly of FtsZ-mts in GUVs, and to observe a very limited initiation of constriction. They almost completely ignore early and recent studies that have previously achieved results comparable to their FtsZ-mts + FtsN system. Osawa and Erickson 2008 first demonstrated progressive constriction of small diameter tubular liposomes by FtsZ-mts alone; no accessory proteins were needed. Osawa and Erickson 2019 achieved constriction of large GUVs by FtsA* + FtsZ, which in rare instances gave complete septation. Godino and Donelan 2020 showed constriction, by FtsZ + FtsA, of GUVs into daughter vesicles that were separated by only a narrow neck. Godino and Donelan 2022 improved the system by using FtsA. Importantly, they documented that the system could constrict to a tubular neck of 1.0 – 0.5 μm , but never achieved full separation. This is consistent with the observation of Söderström and Daley that in *E. coli* FtsZ disappears at a constriction of 0.2-0.3 μm .

I would suggest that the present mss needs a paragraph in the Introduction, something like the above, calling attention to these prior studies. The Discussion should then have a paragraph on each of these previous studies, with specifics of how the present FtsZ-cytoFtsN system duplicates or improves (I don't think it improves on the FtsZ-FtsA*, but the authors may be able to clarify).

Without going into detail, I think the same approach should be used to relate the present SLB system to previous results from other labs, primarily Loose.

Smaller concerns.

Filament bundles are stated to have an average width 0.4 μm , and minirings a diameter (outside or medium) of 0.37 μm . This is only slightly above the diffraction limit of 0.25 μm , which would be blurred substantially if out of focus. I suggest dropping these measurements unless for a filament in sharp focus, or at least clarifying that the estimated width is only slightly above the diffraction limit.

In Fig. 2e we are shown a highly elliptical ring in projection. But is this really elliptical, or is it circular and tilted. If so a reference to d_{minor} is meaningless.

“Previous reconstitution efforts often required macromolecular crowding agents, such as Ficoll and dextran, to promote FtsZ bundling and initial membrane constriction within vesicles (refs8,40,41)... In contrast, we here demonstrate the robust FtsZ-driven membrane constriction without the use of crowding agents, relying solely on cytoFtsN-mediated division ring stabilization. Our findings highlight the intrinsic force-generating capacity of FtsZ.”

Kohyama and Schwille did use crowding agents to achieve functional Z rings in liposomes. But this is not universal. Osawa and Erickson, and also Godino and Danelon achieved functional Z rings without crowders. It might be simplest just to remove any mention of crowders. Otherwise it would be important to address why they were needed in some but not other constructions.

“Once such a ring is established, reducing filament dynamics, achieved here via cytoFtsN-mediated GTPase suppression, is essential to allow the system to stabilize, reorganize, and exert contractile forces at the micrometer scale.” This is a speculation, and one that I doubt. Reduced GTPase is probably a secondary result of bundle formation, which reduces the rate of subunit exchange. I think it more likely that the primary effect of the bundling is to force longer protofilaments and prevent their collapse into the ~0.5 - 1 μm rings.

“The addition of cytoFtsN significantly prolonged FRAP recovery half-time.” The data to me look like a small but measurable increase in recovery time. “The slower turnover of FtsZ subunits induced by cytoFtsN suggests that the peptide affects monomers rather than higher-ordered structures.” How could the peptide affect monomers? I would suggest that the slower recovery means that the average filament length is somewhat longer with FtsN.

A technical question - In Fig. 4b the anisotropy is apparently measuring fluorescent FtsN. But what is measured in 4c, where FtsN varies from 0 to 100 μM ? Is this turbidity?

Finally, a suggestion for a new experiment, which could be very interesting. The cytoFtsN is essentially a cationic peptide, which alphaFold suggests has a tendency to make an alpha helix. I would suggest that the authors try a generic cationic peptide like poly lysine.

Some additional problems with references, and suggestions.

Ref 14 is not even mentioned in the text. It is the Yang..Xiao treadmilling in *Bacillus*. If that paper is mentioned you should also include the companion paper of Bisson..Garner showing treadmilling in *E. coli*. Ref 15 is to the de Boer 1992 original discovery of FtsZ. If that paper is mentioned you should also mention the co-discoveries by RayChaudhury and Park, and Mukherjee and Lutkenhaus. However, it might be better to skip these historic discoveries and go directly to treadmilling, which GTP powers. You include several references to in vitro studies of treadmilling. I think it would be useful to add references to three papers that have presented models to explain how treadmilling works: Du and Lutkenhaus 2018, Wagstaff and Löwe 2017, Corbin and Erickson 2020.

Line 70 has ref 31, and line 74 jumps to refs 46,46. At least some of the intervening refs are found in Discussion, but I think the journal will want you to order them as mentioned.

Reviewer #2

(Remarks to the Author)

In this manuscript, Schwille and co-workers present a simplified reconstitution system to mimic Z-ring-directed division. By employing giant unilamellar vesicles (GUVs) and incorporating the proteins cytoFtsN and FtsZ-366-mts, they successfully construct a functional divisome-like ring. This minimal system is a significant reduction in complexity compared to the natural, multi-protein machinery. The authors provide a systematic investigation into the role of cytoFtsN, convincingly demonstrating that electrostatic interactions between FtsZ and cytoFtsN are crucial for FtsZ bundle formation and ring stability. Overall, this is an excellent, well-written study with compelling results that is suitable for publication in *Nature Communications* after the following concerns are addressed:

1. The authors state, “Rather than using full-length transmembrane FtsN, we focused on its cytoplasmic tail cytoFtsN.” What was the specific rationale for this choice? Would the full-length transmembrane FtsN function similarly or perhaps more effectively in this reconstituted system?

2. The authors emphasize that their system requires no exogenous crowding agents. However, the concentration of cytoFtsN (120 μM) is exceptionally high compared to that of FtsZ-mts (3 μM). Could cytoFtsN itself be acting as a molecular crowder to promote bundle formation? If its primary role is purely through specific electrostatic binding, why are such high concentrations necessary? Complementary experiments using traditional crowders like Ficoll or Dextran at varying ratios of FtsZ-mts to cytoFtsN could help distinguish specific binding effects from non-specific crowding.

3. Figure 2g and Supplementary Figure 2 show impressive vesicle constriction and division (almost) events. It would be strengthened by a more detailed discussion of the physical mechanism driving this process. Is constriction powered by the GTP hydrolysis-dependent contraction of the FtsZ/cytoFtsN ring, or by the rearrangement of the membrane-attached filaments? A deeper explanation of the proposed chemical or physical principles would be valuable for the reader.

4. The authors note that while Min protein oscillations achieved transient central positioning of the Z-ring, rings often self-positioned correctly even in their absence. This suggests the Min system is not strictly necessary for positioning in this

reconstitution. Could the authors elaborate on why the Min system appears less influential here than in vivo? Is this due to the simplicity of the system or the inherent self-organization properties of the FtsZ/cytoFtsN ring?

Version 1:

Reviewer comments:

Reviewer #1

(Remarks to the Author)

The authors have constructively addressed the concerns raised in my previous review. I recommend publication.

Reviewer #2

(Remarks to the Author)

The authors have satisfactorily addressed all of my concerns. I have no further questions and recommend the manuscript for publication as it is.

REPLY TO REVIEWERS

We thank the reviewers for the insightful comments and detailed review. Following their insightful suggestions, we performed additional experiments, implemented the recommended corrections, and added new explanations and clarifications throughout the manuscript. Our responses are included in this document immediately following the reviewer's comments. In the revised manuscript, the modified text is highlighted in yellow.

Reviewer #1 (Remarks to the Author):

This paper addresses the question of how to reconstitute FtsZ ring assembly and constriction in large spherical liposomes. They use primarily the construct FtsZ-mts, and show for the first time that adding an excess of cytoFtsN, a 32 aa cationic peptide, enhances the assembly of intact Z rings, which can progressively constrict the liposomes. The GUV constriction seems essentially the same as that reported by Godino and Danelon 2022, using a somewhat different combination, FtsZ (native) plus FtsA*. Although the constriction is very similar, the present result is interesting in showing that there is nothing essential about FtsA*. The generic FtsZ-mts can do the job if provided with a cationic peptide, which probably acts by bundling FtsZ protofilaments. That puts a renewed interest in FtsZ itself as a source of constriction force.

However, the presentation has a major flaw. This is epitomized in the statement (Line 228) "To the best of our knowledge, this is the first large-scale protein-based vesicle transformation process observed in real time." Do the authors really not know the history of related studies? Are they really trying to claim that their FtsZ-FtsN system is the first to achieve "large-scale" constriction?

We thank the reviewer for this valuable comment and apologize for the ambiguity in our original phrasing. Our intention was not to claim priority over previous studies demonstrating FtsZ-driven membrane constriction, but rather to emphasize that, to our knowledge, this is the first instance where a single, stably positioned FtsZ ring, without auxiliary anchors, has been directly observed to constrict a giant vesicle ($> 2.5 \mu\text{m}$ in diameter) continuously over time, from its initial assembly through near-complete transformation into two compartments. We have revised the sentence to clarify this point and to appropriately acknowledge prior work (page 7, line 258-260) as follows:

»This represents a clear demonstration that a single, continuous, membrane-tethered FtsZ ring can accomplish large-scale transformation of a spherical giant vesicle to almost full fission, where the constriction process could be followed in real time.«

1. The authors heavily reference their own 2022 Kohyama and Schwille study for prior studies of reconstitution of FtsZ in liposomes, which used crowding and the Min system to improve the assembly of FtsZ-mts in GUVs, and to observe a very limited initiation of constriction. They almost completely ignore early and recent studies that have previously achieved results comparable to their FtsZ-mts + FtsN system. Osawa and Erickson 2008 first demonstrated progressive constriction of small diameter tubular liposomes by FtsZ-mts alone; no accessory proteins were needed. Osawa and Erickson 2019 achieved constriction of large GUVs by FtsA* + FtsZ, which in rare instances gave complete septation. Godino and Donelan 2020 showed constriction, by FtsZ + FtsA, of GUVs into daughter vesicles that were separated by only a narrow neck. Godino and Donelan 2022 improved the system by using FtsA. Importantly, they documented that the system could constrict to a tubular neck of $1.0 - 0.5 \mu\text{m}$, but never achieved full separation. This is consistent with the observation of Söderström and Daley that in *E. coli* FtsZ disappears at a constriction of $0.2-0.3 \mu\text{m}$.

I would suggest that the present mss needs a paragraph in the Introduction, something like the above, calling attention to these prior studies. The Discussion should then have a paragraph on each of these

52 previous studies, with specifics of how the present FtsZ-cytoFtsN system duplicates or improves (I
don't think it improves on the FtsZ-FtsA*, but the authors may be able to clarify.

*We thank the reviewer for the suggestion and we have accordingly revised the **Introduction** to include*
*a paragraph summarizing the relevant prior work on FtsZ-based membrane constriction, beginning*
*with the stabilizing influence of FtsN and DamX on the constriction zone (page 2, line 69-70) citing*
*Söderström and Daley (2022), as follows:*

*» Recent studies have identified a transmembrane divisome protein FtsN as a key activator of bacterial*
*cell division, which coordinates intracellular division events with cell wall remodeling³⁰. FtsN functions*
*together with its SPOR-domain partner DamX, which contributes to stabilizing the constriction zone*
*during the late stages of cytokinesis³¹«*

Additionally, we also provide more detailed discussion of the key contributions of Osawa and Erickson
(2008, 2013) and Godino et al. (2020, 2022, 2023), highlighting how these studies collectively laid the
foundation for the present work (page 2, lines 87-102), as follows:

*» In contrast, an engineered FtsZ chimera containing membrane-targeting sequence (mts), which*
*enables membrane binding in the absence of native FtsZ anchors ZipA or FtsA, was shown to*
*accumulate at the constricted regions of tubular liposomes (< 2.5 μm), thereby laying the foundation*
*for the minimal divisome model³⁶. However, FtsZ-mts reconstituted within giant unilamellar vesicles (>*
*5 μm in diameter) forms mini-rings and patches that generate only local, concave distortions without*
*inducing large-scale vesicle constriction^{37, 38}. On the contrary, co-reconstitution of FtsZ with the gain-*
*of-function FtsA* in giant vesicles produced dynamic Z-rings capable of constriction and occasional*
*septation, whereas substitution of GTP with slowly hydrolyzing GMPCPP abolished ring formation³⁷.*
*The in situ expression of FtsA and FtsZ within liposomes resulted in their assembly into curved filaments*
*and closed rings, which constricted membranes into extended necks and budding vesicles^{39, 40}.*
*Incorporating the Min system further enabled dynamic FtsZ patterning and equatorial ring positioning,*
*though only slight deformations were achieved^{8, 41}. Together, these advances charted a progressive*
*path toward a minimal synthetic divisome, yet none captured a continuous, time-resolved*
*transformation driven solely by membrane-tethered FtsZ, without auxiliary anchors, in which a single,*
*stably positioned ring progressively reshapes a giant vesicle into two nearly separated daughter*
*compartments. «*

*In the **Discussion**, we now directly compare our findings with these earlier systems, highlighting both*
*the conceptual and methodological distinctions. Specifically, we clarify that in our reconstitution*
*experiments, the FtsZ-cytoFtsN operates without additional protein membrane anchors or spatial*
*regulators. Yet it produces a single, centrally positioned FtsZ ring capable of continuous, time-resolved*
*constriction within giant vesicles larger than 2.5 μm in diameter, a scale not achieved in prior*
*reconstitutions. Furthermore, our approach employs a minimal peptide regulator that simultaneously*
*promotes filament alignment and stability.*

*We have integrated these points into the revised Discussion (page 14, line 486-497) to first summarize*
*prior studies, as follows:*

*» FtsZ plays a central role in driving membrane invagination, yet its mechanical sufficiency in isolation*
*has been debated. While pioneering studies established that FtsZ-Venus-mts alone can assemble rings*
*to constrict narrow tubular liposome (<2.5 μm), its inability to constrict larger GUVs forming only local*
*distortions suggested a requirement for auxiliary anchors like FtsA or ZipA³⁶. Indeed, co-reconstituting*
*FtsZ with the gain-of-function FtsA*, or expressing wild-type FtsA and FtsZ in cell-free systems, yielded*
*dynamic rings and budding necks^{37, 38, 40}. These FtsA-FtsZ assemblies frequently formed clusters that*

constricted the membrane into budding necks, yet under the studied conditions, they were insufficient
to complete abscission. In contrast, by utilizing FtsZ-Venus-mts and cytoFtsN, we demonstrate here
that FtsZ-mts alone is sufficient to drive global constriction in giant vesicles. This overcomes limitations
observed in previous multicomponent systems, such as the cell-free expression of FtsA, FtsZ, and
MinCDE, which resulted in equatorially centered but diffuse rings capable of only shallow membrane
indentations⁸. «

We then discuss how our system extends and refines these approaches (**page 14, line 525-540**):

» In contrast to previous studies^{8, 37, 39, 40-41}, our system represents a conceptual and methodological
advance in reconstituting FtsZ-ring driven constriction at the scale of giant vesicles. Rather than relying
on multiple recombinant proteins such as FtsA or its hyperactive variant FtsA*, or on cell-free
expression systems requiring continuous protein synthesis, we used a minimal two-component setup
consisting of FtsZ-366-mts and a short cytoplasmic peptide of FtsN. Our findings demonstrate that
large-scale constriction of giant vesicles can be achieved by a single FtsZ ring structure alone, when
combined with a regulatory peptide that enhances its dynamic self-organization. This minimal
configuration eliminates the need for accessory membrane anchors while preserving functional
coupling between FtsZ dynamics and membrane remodeling. Remarkably, under these conditions we
observed large-scale constriction of giant vesicles (>2.5 μm in diameter) initiated from a single,
centrally positioned FtsZ ring that progressively tightened over time until near-complete vesicle
division, capturing the full transformation process rather than only its intermediates or budded final
stages. The use of cytoFtsN as a small, diffusible modulator provides a new strategy for tuning filament
alignment, stability, and constriction dynamics without requiring stoichiometric assembly of larger
accessory proteins, thereby significantly simplifying and stabilizing the minimal divisome
reconstitution. «

Without going into detail, I think the same approach should be used to relate the present SLB system
to previous results from other labs, primarily Loose.

We have accordingly expanded the relevant section of the **Introduction** with previous supported lipid
bilayer (SLB) reconstitutions, primarily from the groups of Loose and Löwe (**page 2, line 71-80**). These
studies are now briefly discussed to place our SLB results in the context of their pioneering work on
cytoFtsN-FtsA-FtsZ filament dynamics and membrane organization, as follows:

» In vitro and structural studies have provided detailed insight into how cytoFtsN modulates FtsA and
FtsZ dynamics at the onset of division. In reconstituted systems, FtsA couples FtsZ treadmilling to the
membrane and recruits cytoFtsN, which is captured by FtsA and co-migrates with FtsZ-FtsA filaments,
demonstrating a direct cytoplasmic connection between the early FtsZ-ring scaffold and downstream
division factors^{33, 34}. Structural analyses further revealed that binding of cytoFtsN to FtsA induces a
conformational transition from single to antiparallel double filaments, increasing curvature sensitivity
and stabilizing the proto-ring on negatively curved membranes³⁵. Together, these findings establish
cytoFtsN as an allosteric regulator that reorganizes FtsA filaments to strengthen their coupling with
FtsZ, thereby priming the divisome for the initiation of constriction. «

Smaller concerns.

Filament bundles are stated to have an average width 0.4 μm, and minirings a diameter (outside or
medium) of 0.37 μm. This is only slightly above the diffraction limit of 0.25 μm, which would be blurred
substantially if out of focus. I suggest dropping these measurements unless for a filament in sharp
focus, or at least clarifying that the estimated width is only slightly above the diffraction limit.

We agree that the measured filament bundle width ($\sim 0.4 \mu\text{m}$) and mini-ring diameter ($\sim 0.37 \mu\text{m}$) are
close to the diffraction limit of our confocal imaging setup. To avoid potential over interpretation, we
have removed the quantitative data referring to the apparent filament bundle width in solution and
the FtsZ ring inside GUVs.

For the mini-ring diameter, we have revised the text in the **Results** section to clarify that these values
represent apparent, diffraction-limited measurements rather than absolute structural dimensions
(page 6, line 248-252), as follows:

» In contrast, encapsulation of FtsZ-366-mts alone resulted in the formation of significantly smaller
rings with an estimated **apparent** diameter of $\sim 0.37 \mu\text{m}$, **which is near the diffraction limit of our**
**imaging system**, and these structures failed to induce membrane deformations beyond the initial ring
dimensions. «

We also emphasize that these measurements were obtained from in-focus filaments and are presented
solely for comparative purposes, providing an approximate indication of diameter variation rather
than precise quantitative dimensions.

In Fig. 2e we are shown a highly elliptical ring in projection. But is this really elliptical, or is it circular
and tilted. If so a reference to D_{minor} is meaningless.

The apparent ellipticity of the ring in Fig. 2e indeed results from a three-dimensional circular ring that
is tilted relative to the imaging plane, as confirmed by z-stack inspection. We have clarified this in the
Figure legend and removed the reference to D_{minor} , since the projection of a tilted circular ring can
appear elliptical in two dimensions. The text now specifies that the structure represents a 3D circular
ring viewed at an angle (page 8, line 281-283; Fig. 2e), as follows:

»In the presence of cytoFtsN, 3D projections reveal a representative FtsZ-ring in the equatorial plane
($D=10.93 \mu\text{m}$), with an elliptical shape resulting from a three-dimensional circular ring that is tilted
relative to the imaging plane. «

“Previous reconstitution efforts often required macromolecular crowding agents, such as Ficoll and
dextran, to promote FtsZ bundling and initial membrane constriction within vesicles (refs8,40,41)... In
contrast, we here demonstrate the robust FtsZ-driven membrane constriction without the use of
crowding agents, relying solely on cytoFtsN-mediated division ring stabilization. Our findings highlight
the intrinsic force-generating capacity of FtsZ.” Kohyama and Schwille did use crowding agents to
achieve functional Z rings in liposomes. But this is not universal. Osawa and Erickson, and also Godino
and Danelon achieved functional Z rings without crowders. It might be simplest just to remove any
mention of crowders. Otherwise it would be important to address why they were needed in some but
not other constructions.

We thank the reviewer for this insightful comment. While macromolecular crowders like Ficoll or
Dextran are well known to enhance later interactions among FtsZ filaments, thereby promoting FtsZ
bundling and ring formation within GUVs (González et al., 2003, J. Biol. Chem.), their use indeed is not
universally required. Notably, Osawa and Erickson (PNAS, 2013), demonstrated the formation of
functional FtsZ rings in submicron-diameter tubular liposomes and GUVs without crowding agents.
This observation suggests that spatial confinement within narrow geometries facilitates functional ring
assembly. Furthermore, in situ protein expression within GUVs has also been shown to yield visible FtsZ
bundles even in the absence of molecular crowders (Godino and Danelon, Adv. Biol., 2023). However,
in reconstitutions of recombinant proteins encapsulated into GUVs by the double-emulsion transfer

*method, co-encapsulation of FtsA and FtsZ without crowding agents often fails to yield membrane-*
*associated FtsZ structures, possibly due to a low encapsulation efficiency of FtsA.*

*Our findings therefore add a new perspective to this assay. We show that molar excess of cytoFtsN*
*promotes the formation of large FtsZ rings and drives membrane constriction also in GUVs without any*
*crowding agents. Importantly, the concentration of cytoFtsN required is well below levels at which*
*excluded-volume effects comparable to Ficoll would occur, ruling out crowding as the mechanism.*
*Instead, cytoFtsN likely promotes ring formation through direct, nonspecific interactions with FtsZ,*
*providing a distinct molecular route to force generation.*

*We have revised the **Results** section, **page 6, lines 213-219** to clarify this point, explicitly distinguishing*
*between the effects of crowding and confinement on FtsZ assembly and stating that our experiments*
*were designed to use cytoFtsN under crowding-free conditions to enable more efficient constriction.*

*» Previous reconstitution studies combining FtsZ, FtsA and Min proteins often relied on molecular*
*crowding agents to enhance FtsZ bundling and facilitate the formation and equatorial positioning of*
*large rings in giant vesicles⁸, though such agents were also reported to hinder efficient constriction^{36,}*
*⁴². Notably, crowders are not universally required, as confinement in narrow geometries can also*
*support functional small-scale ring assembly^{36,42}. Here, we sought to stabilize FtsZ bundles and division*
*rings using only cytoFtsN, without the need for crowding agents. «*

*“Once such a ring is established, reducing filament dynamics, achieved here via cytoFtsN-mediated*
*GTPase suppression, is essential to allow the system to stabilize, reorganize, and exert contractile*
*forces at the micrometer scale.” This is a speculation, and one that I doubt. Reduced GTPase is*
*probably a secondary result of bundle formation, which reduces the rate of subunit exchange. I think*
*it more likely that the primary effect of the bundling is to force longer protofilaments and prevent*
*their collapse into the ~0.5 - 1 μm rings.*

*We thank the reviewer for their insightful mechanistic comment and for identifying an imprecise*
*statement in our manuscript. We agree that the primary role of cytoFtsN is architectural, and that*
*reduction in GTPase activity is likely a secondary consequence of physical stability accomplished by*
*bundling. The hypothesis that the filaments stabilization serves to maintain longer and non-curved*
*protofilaments and prevent their collapse into smaller, non-productive rings is strongly supported by*
*our data. As we show on supported lipid bilayers, cytoFtsN promotes the formation of straightened*
*and aligned filaments (**page 10, Fig. 3a, c**). Furthermore, as described in the manuscript (**page 6, lines***
***248-252** and shown in **Fig. 1c and 2e**, the FtsZ rings formed in the presence of cytoFtsN are significantly*
*larger and are the only ones capable of large-scale membrane deformations, confirming their*
*enhanced functionality.*

*As the reviewer points out, the bundles stabilization likely reduces filament dynamics. However, in our*
*system, the baseline filament dynamics was first globally reduced by the inclusion of the slowly-*
*hydrolyzable GTP analog, GMPCPP. As we note in the manuscript (**page 6, lines 209-213**), this step was*
*essential to prevent the rapid polymerization and aggregation that occurred with GTP alone.*

*Therefore, our refined model posits a two-step process for the formation of a contractile ring: GMPCPP*
*provides the necessary baseline filaments stability, while cytoFtsN provides the crucial architectural*
*stabilization. We note that such filament stabilization appears to be a prerequisite for maintaining*
*large-scale ring integrity and force transmission in the confined GUV geometry.*

To correct ambiguity in our statement, we have rewritten the relevant section of the manuscript. The
revised text (**page 15, lines 574-579 and 581-583** highlight in yellow) with more accurate presentation
of the of events and greater mechanistic clarity is as follows:

» Once such a ring is established, the stabilization of its constituent filaments is essential for the system
to reorganize and progressively constrict over a many-micrometer scale. In our system, this stability is
achieved through a two-step process: first, by globally reducing filament turnover with the addition of
GTP analog GMPCPP together with GTP, and second, by the architectural action of cytoFtsN, which
constrains the filaments into a robust, higher-order structure. «

» We hypothesize that the stabilized FtsZ-cytoFtsN ring then constricts through a finely-tuned
combination of both, filament shortening and turnover, triggered by the gradual depletion of GTP
within the vesicle. «

“The addition of cytoFtsN significantly prolonged FRAP recovery half-time.” The data to me look like a
small but measureable increase in recovery time. “The slower turnover of FtsZ subunits induced by
cytoFtsN suggests that the peptide affects monomers rather than higher-ordered structures.” How
could the peptide affect monomers? I would suggest that the slower recovery means that the average
filament length is somewhat longer with FtsN.

We thank the reviewer for careful analysis of our FRAP data and for providing a more likely
interpretation of the results. We agree with both points raised. First, we have revised the text to more
accurately describe the effect of cytoFtsN on FRAP recovery as a modest prolongation (**page 9, line**
**338**). Second, upon re-evaluating our data, we agree that the slower recovery reflects an increase in the
average length and stability of the filaments. Therefore, we corrected the relevant section of the
manuscript (**page 9, lines 339-340**):

» The slower turnover of FtsZ subunits induced by cytoFtsN likely reflects the increased size and stability
of the filaments. «

which now reflect better that the slower FRAP recovery is a direct reflection of cytoFtsN’s architectural
role in stabilizing FtsZ filaments into larger bundled structure.

A technical question - In Fig. 4b the anisotropy is apparently measuring fluorescent FtsN. But what is
measured in 4c, where FtsN varies from 0 to 100 μM ? Is this turbidity?

In figure 4c, where we follow filament depolymerization kinetics, we used 10 μM wild type FtsZ
together with tracer amount (50 nM) of Alexa-488-labelled wild type FtsZ. We have now added this
clarification to the figure caption (**page 12, line 425**), and the same information in **Methods** section
(**page 18, lines 737, and 742**).

Finally, a suggestion for a new experiment, which could be very interesting. The cytoFtsN is essentially
a cationic peptide, which alphaFold suggests has a tendency to make an alpha helix. I would suggest
that the authors try a generic cationic peptide like poly lysine.

To address the very interesting question whether the observed rearrangement of FtsZ network is a
specific effect of cytoFtsN or whether it can be recapitulated by a generic polycation, we have
performed new experiments with cationic polymer Poly-L-Lysine (PLL).

To establish a functionally relevant concentration of PLL, we calculated its molar concentration
required to provide the same total positive charge as our maximal concentration of cytoFtsN (120 μM).

*CytoFtsN* ($M_w = 3.6$ kDa) has a net charge of +10 at physiological pH (7.4). The PLL we used here has a
 molecular weight range 4-15 kDa, corresponding to an average charge of approximately +74 per
 molecule. A charge-equivalency calculation indicated that 16 μ M PLL provides a total positive charge
 comparable to 120 μ M. We tested the effect of PLL on FtsZ-mts filament architecture using TIRF
 microscopy and the protocol described in manuscript's methods section. At the charge-equivalent
 concentration, PLL had disruptive effect, preventing the formation of an ordered FtsZ filament network.
 We therefore tested a range of lower concentrations. While 100-fold lower concentration (0.16 μ M)
 induced disordered filament aggregation, a concentration of 0.032 μ M had no observable effect on FtsZ
 filament network organization (Figure R-1). This could be explained either by specific sequence or
 charge-distribution effects within cytoFtsN, or by the possibility that the high positive charge density
 of PLL disrupts proper filament organization.

**Figure R-1 | TIRF microscopy images showing the effect of a generic polycation, Poly-L-Lysine (PLL) on FtsZ**
 **filament architecture.** 3 μ M FtsZ-Venus-mts in the presence of decreasing concentrations of PLL indicated above
 each image. Unlike cytoFtsN, which organizes filaments into straightened bundles, PLL either prevents filament
 network formation (at 16 μ M) or induces disordered aggregates at lower concentrations (at 0.16 μ M).

 Some additional problems with references, and suggestions. Ref 14 is not even mentioned in the text.

We thank the reviewer for noticing this oversight. We apologize for the omission, reference 14 was not
 cited in the main text. This has now been corrected, and the reference has been appropriately included
 in the revised manuscript (page 2, line 52).

It is the Yang..Xiao treadmilling in Bacillus. If that paper is mentioned you should also include the
 companion paper of Bisson.. Garner showing treadmilling in E coli. Ref 15 is to the de Boer 1992
 original discovery of FtsZ. If that paper is mentioned you should also mention the co-discoveries by
 RayChaudhury and Park, and Mukherjee and Lutkenhaus. However, it might be better to skip these
 historic discoveries and go directly to treadmilling, which GTP powers. You include several references
 to in vitro studies of treadmilling. I think it would be useful to add references to three papers that have
 presented models to explain how treadmilling works: Du and Lutkenhaus 2018, Wagstaff and Löwe
 2017, Corbin and Erickson 2020.

We have revised the text accordingly. To streamline the introduction, we have omitted the early
 historical (de Boer 1992) and focus instead on GTP-driven treadmilling as the key functional feature of
 FtsZ dynamics. In addition, we now reference the three theoretical and modeling studies (Du and
 Lutkenhaus 2017, Mol Microbiol., Wagstaff and Löwe, 2017, Nat Rev Microbiol, Corbin and
 Schumacher et al., 2020, Acta Crystallogr F Struct Biol Commun.) that provide a mechanistic
 explanation of directional filament dynamics (page 2, line 53-57). The revised paragraph now reads:

» FtsZ treadmilling is a GTP hydrolysis driven process in which filaments continuously polymerize at
 one end and depolymerize at the other through a cooperative conformational switch, producing
 directional, circumferential motion around the division site that organizes and guides the movement
 of peptidoglycan synthases FtsW and FtsI to ensure uniform septal wall synthesis during cytokines^{15,}
 345 ^{16, 17.}«

Line 70 has ref 31, and line 74 jumps to refs 46,46. At least some of the intervening refs are found in Discussion, but I think the journal will want you to order them as mentioned.

The issues with the referencing order were also now corrected and all citations now appear in sequential order.

Reviewer #2 (Remarks to the Author):

In this manuscript, Schwille and co-workers present a simplified reconstitution system to mimic Z-ring-directed division. By employing giant unilamellar vesicles (GUVs) and incorporating the proteins cytoFtsN and FtsZ-366-mts, they successfully construct a functional divisome-like ring. This minimal system is a significant reduction in complexity compared to the natural, multi-protein machinery. The authors provide a systematic investigation into the role of cytoFtsN, convincingly demonstrating that electrostatic interactions between FtsZ and cytoFtsN are crucial for FtsZ bundle formation and ring stability. Overall, this is an excellent, well-written study with compelling results that is suitable for publication in Nature Communications after the following concerns are addressed:

1. The authors state, "Rather than using full-length transmembrane FtsN, we focused on its cytoplasmic tail cytoFtsN." What was the specific rationale for this choice? Would the full-length transmembrane FtsN function similarly or perhaps more effectively in this reconstituted system?

We thank the reviewer for this insightful comment. We agree that assessing whether full-length or membrane-anchored FtsN could act more effectively in our reconstituted system is an important question. Our initial rationale for using only the cytoplasmic tail (cytoFtsN) was to specifically investigate its direct interaction with FtsZ (and FtsA), while avoiding the need for the periplasmic and transmembrane domains reconstitution and proper orientation in the membrane. Nevertheless, motivated by the reviewer's suggestion, we performed additional experiments using an extended variant of cytoFtsN that included its native transmembrane domain:

>transFtsN
MAQRDYVRRSQPAPSRKKKSTSRKKQRNLPAVSPAMVAIAAAVLVTFIGGLYFI

We hypothesized that such an anchor might enhance membrane coupling and promote stronger constriction. However, this construct led to severe lipid–protein aggregation, preventing the formation of intact GUVs (Fig. R-2a, left panel).

Next, to test the role of membrane association in a more controlled manner, we used a His-tagged cytoFtsN (cytoFtsN-His) and anchored it to the membrane of GUVs or SLBs containing NTA lipids (specifically, 1, 2- dioleoyl- sn- glycerol- 3- [(N- (5- amino -1- carboxypentyl) iminodiacetic acid) succinyl] (nickel salt), DOGS-Ni-NTA). In GUVs, membrane-anchored cytoFtsN-His failed to promote the formation of coherent large-scale FtsZ ring (Fig. R-2a, middle panel). Furthermore, increasing the concentration of the anchored peptide was detrimental, causing a significant decrease in vesicle yield (Fig. R-2b), which was higher if more peptide was attached to the membrane of GUVs (higher DOGS-Ni-NTA concentrations).

To understand the underlying mechanism, we performed parallel experiments on SLBs using TIRF microscopy (Fig. R-2, c). We found that cytoFtsN-His when anchored to the lipid bilayer (after unbound peptide was removed by washing), failed to induce filament bundling or reorganization (Fig. R-2 c, middle panel). Similar results were obtained with increasing concentrations of DOGS-Ni-NTA (from 1% to 10 %), which enabled attachment of more peptide to the SLBs. To exclude that the His tag affects

the peptide activity in any way, we also tested non-anchored cytoFtsN-His using SLBs without DOGS-
 Ni-NTA. In contrast to anchored, non-anchored cytoFtsN-His behaved similarly as the untagged
 cytoFtsN, efficiently straightening FtsZ filaments even at lower concentration (30 μ M) than non-tagged
 cytoFtsN (Fig. R-2, d), probably due to increase positive charges of histidine residues and consequent
 better interaction to FtsZ. Importantly, our findings are in agreement with the *in vitro* reconstitution
 by Baranova et al. (Nat Microbiol, 2020), where they demonstrated that a membrane-anchored FtsN
 was unable to recruit FtsZ filaments in absence of FtsZ's native anchor, FtsA. Additionally, they report
 that the non-attached cytoFtsN-His does not affect the FtsZ-FtsA pattern and shows only weak binding
 to FtsZ-FtsA filaments. However, it is critical to note that in that case concentration of cytoFtsN was in
 the sub-stoichiometric ratio (a 3-fold lower molar concentration of cytoFtsN relative to FtsZ), a
 condition designed to probe for higher-affinity recruitment. Our results are therefore in agreement
 with the findings of Baranova et al. as we show that while soluble cytoFtsN does not affect FtsZ
 architecture network at low concentrations, it possesses a concentration-dependent ability to stabilize
 and straighten the FtsZ filaments if it is in molecular excess.

The loss of the ability of cytoFtsN to reorganize FtsZ filaments upon membrane anchoring in our system
 points to two plausible reasons. First, the two-dimensional nature of the membrane may prevent the
 cytoFtsN from achieving the high local concentration required to drive the lower-affinity interactions
 we observe in the three-dimensional solution. Second, tethering the peptide to the membrane surface
 likely imposes steric hindrance and restricts its conformational freedom. This would prevent the
 cytoFtsN from effectively bridging multiple FtsZ filaments, an action essential for its architectural
 function.

**Figure R-2| Membrane anchoring of cytoFtsN impairs GUV formation, but not the FtsZ-ring organization. a**
 Reconstitution of 5 μ M FtsZ-366-mts (green) with different cytoFtsN variants (100 μ M) extended with its native
 transmembrane domain caused extensive lipid-protein aggregation, preventing the formation of intact vesicles
 (left panel). His-tagged cytoFtsN (His-cytoFtsN) tethered to GUV membranes containing 1 mol% Ni-NTA-DOGS
 (middle panel). Soluble (non-anchored) cytoFtsN supported the assembly of continuous, large-scale FtsZ-mts
 mesh architectures (right panel). **b** Quantification of encapsulation efficiency at increasing cytoFtsN-His, shown
 as differential vesicles yields at 1 mol% and 5 mol% Ni-NTA-DOGS lipid concentrations. Data represent the
 average of four 1 mm² tile scans per condition. **c** Representative TIRF microscopy images (n = 3) showing the
 architecture of FtsZ-mts filament patterns (green) assembled on supported lipid bilayers (SLBs) containing 1 mol%
 Ni-NTA-DOGS. Images were acquired 10 min after the addition of 4 mM GTP, in the absence (left) or presence
 (middle and right) of 120 μ M cytoFtsN-His. Unattached cytoFtsN-His was either removed by washing with
 reaction buffer (removed unattached peptide) or retained in the reaction chamber (not removed unattached

peptide). **d** Representative TIRF images showing the spatial organization of FtsZ filaments (3 μM) on SLBs lacking
Ni-NTA-DOGS, in the presence of increasing concentrations of cytoFtsN-His (30, 60, and 120 μM). Scale bar: 5
436 μm .

2. The authors emphasize that their system requires no exogenous crowding agents. However, the
concentration of cytoFtsN (120 μM) is exceptionally high compared to that of FtsZ-mts (3 μM). Could
cytoFtsN itself be acting as a molecular crowder to promote bundle formation? If its primary role is
purely through specific electrostatic binding, why are such high concentrations necessary?
Complementary experiments using traditional crowders like Ficoll or Dextran at varying ratios of FtsZ-
mts to cytoFtsN could help distinguish specific binding effects from non-specific crowding.

We thank the reviewer for insightful comment and pointing out that the high concentrations of
cytoFtsN (120 μM) could potentially mediate its effect through non-specific molecular crowding.
However, the concentration of cytoFtsN used in our experiments is less than 0.5 g/L (molar mass
~ 3000), well below levels where excluded-volume effects similar to Ficoll crowding would occur
(Minton, 1998, *Methods Enzymol.*). To estimate the magnitude of excluded-volume effects by
cytoFtsN we calculated the fractional volume occupancy (φ) using the relationship $\varphi = vc$, where v is
the partial specific volume (cm^3/g) and c is the mass concentration of the molecule (Hall and Minton,
2003, *Biochim Biophys Acta.*). As cytoFtsN is positively charged peptide, and not typical globular
protein, we calculated a specific partial volume based on its amino acid composition (MAQRD YVRRS
QPAPS RRRKS TSRKK QRNLP AV). The specific partial volume calculated as the ratio between volume
of each residue (v_i), and is the mass of that residue M_i ($\bar{v} = (\sum n_i v_i) / (\sum n_i M_i)$), resulted in 0.948 cm^3/g ,
leading to the calculation of the volume occupancy of cytoFtsN at its maximal tested concentration of
120 μM (0.456 g/L):

$$\varphi (\%) = 0.948 \text{ cm}^3/\text{g} \times 0.45 \times 10^{-6} \text{ g}/\text{cm}^3 * 100 = \mathbf{0,04266 \%}.$$

Macromolecular concentrations in the cytoplasm of *E. coli* and eukaryotic organelles are estimated to
occupy 5–40 % of the cellular volume and therefore typically used in studies dealing with crowding
(Alfano et al., 2024, *Chem Rev.*, Speer et al., 2022, *Annu Rev Biophys.*). Therefore, the calculated
volume occupancy of cytoFtsN rules out crowding as the mechanism of activity of cytoFtsN.
Additionally, the FtsZ ring formation driven by highly charged cytoFtsN resembles cation-induced
protein bundling via electrostatic interactions, where cations bind to protein surfaces. In our case, the
effect of cytoFtsN on FtsZ filaments network occurs at a high molar excess of peptide, indicating
unsaturable binding rather than site-specific interactions. Similar phenomena have been observed in
other biopolymers, where high cation levels promote actin bundling and DNA condensation
(Castaneda et al., 2021, *Front. Phys.*).

While the calculation strongly suggested a negligible crowding effect of cytoFtsN, we proceeded to test
this experimentally. First, we used TIRF microscopy to test the direct effect of the Ficoll 70 on FtsZ
filaments architecture. We found that concentration of Ficoll 70 matching the volume occupancy of
120 μM cytoFtsN or a 2x higher, had no observable effect on the FtsZ filament network architecture
(Fig. R. 3a).

Further, we systematically tested a broad range of Ficoll 70 concentrations, from 25 to 150 mg/mL,
which encompasses the range typically used in crowding studies. At 150 mg/mL Ficoll 70 induced
alignment of the FtsZ filaments (in agreement to previously observed phenomenon (González et al.,
2003, *J. Biol. Chem.*), however circular vortices typical for FtsZ architecture on SLBs were also present.
Therefore, the effect was qualitatively different from the effect induced by cytoFtsN (Fig. R. 3b). This
demonstrates that the non-specific crowding cannot fully replicate the architectural rearrangement
induced by cytoFtsN.

Finally, to experimentally separate the effects of cytoFtsN electrostatic binding from non-specific
molecular crowding, we tested concentrations of cytoFtsN which did not induce significant
rearrangements of the FtsZ filament network (namely 3 μM and 30 μM) together with varying
concentrations of Ficoll 70, both on SLBs and within GUVs. In crowding conditions even lower
concentration of cytoFtsN (30 μM) induced FtsZ filament network reorganization (**Fig. R-3, c**).

Next, we tested the effect of Ficoll 70 on FtsZ network architecture in GUVs. Within GUVs, Ficoll 70
induced the formation of transient mesh of FtsZ filaments (**Fig. R-3, d, left panel; Fig. R-3, f**), which
disassembled within 90 minutes. This instability highlights that purely crowding-induced assembly
cannot sustain long-lived FtsZ structures in the absence of stabilizing factors such as cytoFtsN. When
0.8 mM GMPCPP was introduced to slow FtsZ dynamics in crowding conditions, only small, stable mini-
rings incapable of inducing large-scale membrane deformation, were observed (**Fig. R-3, d, middle**
**panel, Fig. R-3, f**). These results demonstrate that generic crowding cannot substitute for cytoFtsN, as
the peptide acts in a more direct manner, modulating FtsZ filament architecture, stability, and
consequently GTPase activity. When 30 μM cytoFtsN was used, FtsZ filaments became more aligned
and long-lived (>90 min), although at lower concentrations (≤ 30 μM) no coherent, centrally positioned
rings were detected (**Fig. R-3, d, right panel, Fig. R-3, f**). The formation of large, organized and stable
FtsZ rings which lead to progressive vesicle constriction required higher cytoFtsN concentrations (**Fig. R-**
**3, e**), indicating that a threshold level of peptide-mediated stabilization is necessary for effective large-
scale membrane remodeling.

In conclusion, molecular crowding alone is insufficient to replicate the stable architectural remodeling
of the FtsZ network induced by cytoFtsN. Instead, our results show a synergistic effect: a crowding
environment promotes a basal level of FtsZ self-organization, but cytoFtsN is needed to stabilize these
transient filaments into long-lived structures and coherent rings capable of driving membrane
deformations.

**Figure R-3 | The effect of Ficoll 70 and cytoFtsN on FtsZ architecture and dynamics.** **a** Comparison of the effect
of *cytoFtsN* and known crowder *Ficoll 70* at the concentration matching the volume occupancy of *cytoFtsN* at
maximal tested concentration 120 μM . Representative micrographs showing the organization of 3 μM FtsZ–
Venus–mts filaments (green) formed either without *cytoFtsN* or crowders (far left), with 120 μM *cytoFtsN* (as
indicated above the image), or with increasing concentrations of Ficoll (0, 65 mg/mL or 1.3 mg/mL as indicated),
10 min after 2 mM GTP addition. **b** Representative micrographs showing the organization of 3 μM FtsZ–Venus–
mts filaments formed with increasing crowding concentrations of Ficoll 70. **c** Combined effect of increasing
concentration of Ficoll 70 and *cytoFtsN* on FtsZ architecture on SLBs. TIRF microscopy images showing the
organization of 3 μM FtsZ–Venus–mts filaments formed with 30 μM *cytoFtsN* under increasing concentrations
of Ficoll 70, 10 min after GTP addition. **d** Representative confocal images showing the FtsZ structures formed
within GUVs in presence of Ficoll 70 (100 mg/mL) over time under different conditions: with or without GMPCPP
and with or without 30 μM *cytoFtsN*. **e** Representative image of a stable, centered FtsZ ring formed in the
absence of Ficoll 70 but in the presence of 100 μM *cytoFtsN*. **f** Quantification of FtsZ structures inside GUVs ($n =$
100). In the presence of Ficoll 70 alone, predominantly FtsZ mesh were observed (1); the addition of GMPCPP

(0.8 mM) led to the appearance of mini-rings (2); combining GMPCPP with cytoFtsN (30 μ M) produced mixed
mesh and mini-ring architectures (3); while cytoFtsN alone (30 μ M, no Ficoll 70) primarily yielded mesh-like
networks (4). Scale bars, 5 μ m.

3. Figure 2g and Supplementary Figure 2 show impressive vesicle constriction and division (almost)
events. It would be strengthened by a more detailed discussion of the physical mechanism driving this
process. Is constriction powered by the GTP hydrolysis-dependent contraction of the FtsZ/cytoFtsN
ring, or by the rearrangement of the membrane-attached filaments? A deeper explanation of the
proposed chemical or physical principles would be valuable for the reader.

*The question of how, and whether at all, a measurable contractile force is generated in our system is*
*indeed a critical one, and we agree that a more detailed discussion would be valuable. However,*
*decisive experiments will be complicated and require significantly more time and resources. We*
*therefore decided to be very careful with claims about actual physical forces, and tried to avoid the*
*term in our revised manuscript. What can, on the other hand, be clearly and consistently observed, is*
*ring constriction accompanied by large-scale membrane transformation. The least presumptive*
*hypothesis is that ring contraction itself is generated by the shortening of the FtsZ filaments due to*
*eventual gradual GTP depletion within GUVs, while cytoFtsN maintains ring continuity, allowing for*
*the whole structure to contract as a single, cohesive unit. A mixture of GTP/GMPCPP is needed to slow*
*FtsZ turnover, allowing filaments to anneal, align, and persist long enough to organize into coherent*
*ring structures. Additionally, cytoFtsN stabilizes filaments into long, coherent bundles that can span*
*the GUV surface. Once a continuous FtsZ–cytoFtsN ring is established, its gradual reduction in diameter*
*likely results from slow filament depolymerization and reorganization driven by progressive GTP*
*depletion within the vesicle. As GTP is consumed, the balance between polymerization and*
*depolymerization shifts towards disassembly of the ring. Because the filaments are tethered to the*
*membrane stabilized by cytoFtsN, local shortening of filaments is geometrically converted into inward*
*movement of the ring, effectively tightening the membrane orifice at the constriction site. It should be*
*kept in mind, however, that the whole process takes a significant amount of time (order of hours)*
*comparable to division of live cells (e.g., the much smaller E. coli in rich medium take around 20 min).*
*Since there will always be some volume equilibration along this process, the required force for radial*
*constriction of a giant vesicle may be much lower than intuitively expected, and very hard to model*
*correctly.*

*To provide the potential explanation of the proposed principle, we have incorporated these*
*considerations into the **Discussion** section. Specifically, we added to following sentence (page 15, lines*
**581-583):**

*»We hypothesize that the stabilized FtsZ-cytoFtsN ring then constricts through a finely-tuned*
*combination of both, filament shortening and turnover, triggered by the gradual depletion of GTP*
*within the vesicle.*

4. The authors note that while Min protein oscillations achieved transient central positioning of the Z-
ring, rings often self-positioned correctly even in their absence. This suggests the Min system is not
strictly necessary for positioning in this reconstitution. Could the authors elaborate on why the Min
system appears less influential here than in vivo? Is this due to the simplicity of the system or the
inherent self-organization properties of the FtsZ/cytoFtsN ring?

*In our reconstituted system, Min oscillations facilitated the spatial confinement of FtsZ-cytoFtsN*
*filaments but were not strictly required for equatorial ring positioning. Even in the absence of Min*
*proteins, we frequently observed robust spontaneous midplane localization of FtsZ-cytoFtsN rings. This*
*indicates that the intrinsic self-organization principles of the FtsZ-cytoFtsN network together with the*
*geometric constraints imposed by the vesicle membrane are sufficient to define the division plane. Our*

*aim was also to keep the system minimal, similar in spirit to previous minimal-actin systems that*
*yielded single, equatorial rings (Litschel et al., 2021, Nat commun). We believe that in such reduced*
*environments where the effective persistence length of the filaments is relatively high, membrane*
*curvature becomes the dominant organizing cue. Under these conditions, filaments are probably*
*primarily bent by the vesicle geometry rather than by their intrinsic curvature, naturally biasing the*
*system toward midplane accumulation.*